# A high-resolution synthesis dataset for multistressor analyses along the U.S. West Coast

Esther G. Kennedy[1, 2], Meghan Zulian[1, 2], Sara L. Hamilton[2, 3], Tessa M. Hill[1, 2], Manuel Delgado[2], Carina R. Fish[1, 2], Brian Gaylord[2, 4], Kristy J. Kroeker[5], Hannah M. Palmer[1, 2], Aurora M. Ricart[6, 7], Eric Sanford[2, 4], Ana K. Spalding[8, 9], Melissa Ward[10], Guadalupe Carrasco[11], Meredith Elliott[12], Genece V. Grisby[1], Evan Harris[1], Jaime Jahncke[12], Catherine N. Rocheleau[1], Sebastian Westerink[13], Maddie I. Wilmot[1]

[1]Department of Earth and Planetary Sciences, University of California Davis, Davis, CA, USA
[2]Bodega Marine Laboratory, University of California Davis, Bodega Bay, CA, USA
[3]Oregon Kelp Alliance, Port Orford, OR, USA
[4]Department of Evolution and Ecology, University of California Davis, Davis, CA, USA
[5]Department of Ecology and Evolutionary Biology, University of California Santa Cruz, Santa Cruz, CA, USA
[6]Institut de Ciències del Mar, ICM-CSIC, Barcelona, Spain
[7]Bigelow Laboratory for Ocean Sciences, East Boothbay, ME, USA
[8]Oregon State University, Corvallis, OR, USA
[9]Smithsonian Tropical Research Institute, Panama City, Panama
[10]Coastal and Marine Institute, San Diego State University, San Diego, CA, USA
[11]Department of Biology, Sonoma State University, Rohnert Park, CA, USA
[12]Point Blue Conservation Science, Petaluma, CA, USA
[13]Department of Land, Air and Water Resources, University of California Davis, Davis, CA, USA

*Correspondence to*: Esther G. Kennedy (egkennedy@ucdavis.edu)

**Abstract.** The global trends of ocean warming, deoxygenation, and acidification are not easily extrapolated to coastal environments. Local factors, including intricate hydrodynamics, high primary productivity, freshwater inputs, and pollution, can exacerbate or attenuate global trends and produce complex mosaics of physiologically stressful or favorable conditions for organisms. In the California Current System (CCS), coastal oceanographic monitoring programs document some of this complexity; however, data fragmentation and limited data availability constrain our understanding of when and where intersecting stressful temperatures, carbonate system conditions, and reduced oxygen availability manifest. Here, we undertake a large data synthesis to compile, format, and quality-control publicly available oceanographic data from the U.S. West Coast to create an accessible database for coastal CCS climate risk mapping, available at the National Centers for Environmental Information (Accession 0277984) under the DOI 10.25921/2vve-fh39 (Kennedy et al., 2023). With this synthesis, we combine publicly available observations and data contributed by the author team from synoptic oceanographic cruises, autonomous sensors, and shore samples with relevance to coastal ocean acidification and hypoxia (OAH) risk. This large-scale compilation includes 13.7 million observations from 66 sources and spans from 1949 to 2020. Here, we discuss the quality and composition of the synthesized dataset, the spatial and temporal distribution of available data, and examples of potential analyses. This dataset will provide a valuable tool for scientists supporting policy- and management-relevant

investigations including assessing regional and local climate risk, evaluating the efficacy and completeness of CCS monitoring efforts, and elucidating spatiotemporal scales of coastal oceanographic variability.

# 1 Introduction

Anthropogenic carbon dioxide ($CO_2$) emissions are causing dramatic ocean warming, acidification, and deoxygenation (Caldeira and Wickett, 2003; Doney et al., 2009; Doney 2010; Levitus et al., 2012). Interactions among these stressors can compound the severity of each, often synergistically reducing growth, metabolism, and survival of marine organisms across diverse taxa (e.g., Byrne and Przeslawski, 2013; Gobler and Baumann, 2016). Multiparameter extreme events are increasingly common and destructive (Burger et al., 2013; Breitburg et al., 2015). However, global ocean trends may be masked, modified, or overshadowed in coastal ecosystems by combinations of complex local oceanographic processes, terrestrial runoff, freshwater sources, and high local productivity (Borges and Gypens, 2010; Cai et al., 2011; Fassbender et al., 2011; Frieder et al., 2012; Bauer et al., 2013; Takeshita et al., 2015). Despite thorough documentation of global ocean responses to anthropogenic forcing, understanding more localized conditions in coastal environments, such as the California Current System (CCS), remains an active area of research. Improved understanding of spatiotemporal patterns of warming, deoxygenation, and acidification is key to informing climate resilience and adaptation planning for and by the diverse peoples and ecological communities that depend on the coastal CCS (Field and Francis, 2006; Hodgson et al., 2018; IPCC 2019; Weisberg et al., 2020; Ward et al., 2022).

The CCS is an upwelling system where seasonal winds transport cold, low-oxygen, high-$CO_2$ waters from depth up to nearshore surface environments (e.g,. Hickey, 1979; Huyer, 1983; Chavez and Messié, 2009). Upwelling intensity varies across small spatial and temporal scales and is typically concentrated in the spring and early summer (Hickey, 1979; Marchesiello et al., 2003; Garciá-Reyes and Largier, 2012; Jacox et al., 2018; Cheresh and Fiechter, 2020). During upwelling, extreme values of seasonal dissolved oxygen (DO) and carbonate chemistry parameters such as pH are naturally close to biologically significant thresholds, making organisms in the CCS particularly vulnerable to ocean acidification and hypoxia (OAH) events (e.g., Chan et al., 2008; Connolly et al., 2010; Feely et al., 2008; Gruber et al., 2012; Low et al., 2021; Kekuewa et al., 2022). Local adaptation to high environmental variability may provide some ecological resilience (e.g., Sanford and Kelly, 2011; Kelly and Hofmann., 2013; Donham et al., 2023), but widespread die-offs are already a feature of some OAH events (e.g., Grantham et al., 2004; Barton et al., 2015). The CCS is also vulnerable to warming and heatwaves (Cavole et al., 2016; Frölicher and Laufkötter, 2018; Rogers-Bennett and Catton, 2019; Sanford et al., 2019; Fumo et al., 2020; Cheung and Frölicher, 2020; Free et al., 2023). When extreme temperatures interact with low pH and low oxygen conditions, they can compound the vulnerability of organisms to environmental stressors (e.g., Kroeker et al., 2013; Swiney et al., 2017; Bednaršek et al., 2019; Howard et al., 2020b; Sunday et al., 2021). The balance between local upwelling intensity, warming-induced stratification, and both oceanic and terrestrial influences creates a spatiotemporal mosaic of

coastal ocean conditions which, while previously acknowledged and documented (e.g., Feely et al., 2016a, Chan et al., 2017; Cheresh and Fiechter, 2020), remains incompletely described.


As a result of the connections between upwelling, low oxygen, and acidification events, models predict the CCS's vulnerability to extreme events will increase as climate change progresses (Gruber et al., 2012; Bakun et al., 2015). Relative to a preindustrial baseline, anthropogenic forcing has shallowed the depths of perennially corrosive and hypoxic conditions by more than 50 m (Bograd et al., 2008; Feely et al., 2008; Chan et al., 2008; Gruber et al., 2012). Modeled projections of

the CCS suggest that pH levels are declining sufficiently swiftly that by 2035, the range of annual variability may no longer overlap with conditions present in the 2010s while the calcium carbonate mineral aragonite could be perennially undersaturated at 100 m depth by 2045 (Hauri et al., 2013; Marshall et al., 2017). Meanwhile, nearshore DO content is expected to decline by 10-20 µmol kg$^{-1}$ by the end of the century (Siedlecki et al., 2021). Upwelling-favorable winds may intensify under future warming (Sydeman et al., 2014; Bakun et al., 2015; Wang et al., 2015); although this effect may be

counteracted in some locations by increased stratification of seawater layers (Howard et al., 2020a; Siedlecki et al., 2021) or in areas where wind-driven upwelling is not the dominant process (Garciá-Reyes and Largier, 2010). These competing forces might enhance the disparities between climate hot spots and refugia, underlining the importance of gathering and analyzing climate data with high spatiotemporal resolution.

Despite recognition of the complexity of CCS coastal climate stress, successfully capturing mesoscale, sub-seasonal, and very nearshore patterns of OAH and warming remains challenging. One impediment to unraveling this complexity is the decentralized and non-standardized nature of much OAH monitoring in the CCS, undertaken by governmental, non-profit, and academic centers with varying methodologies and approaches to data accessibility (Taylor-Burns et al., 2020). Further, existing synthesis datasets are not optimized for simultaneous analysis of nearshore warming, deoxygenation, and

acidification risks (e.g., Hofmann et al., 2011; Sharp et al., 2022). For chemical oceanographers and modelers, the Surface Ocean CO$_2$ Atlas (SOCAT, Sabine et al., 2013; Bakker et al., 2016) and Coastal Ocean Data Analysis Product in North America (CODAP-NA, Jiang et al., 2021) are also valuable resources. However, the former includes only surface seawater observations of one principal parameter of the carbonate system, while the latter includes only discrete bottle observations from oceanographic cruises while excluding autonomous sensor observations and shore samples. SOCAT and CODAP-NA

are high-quality and extremely well-curated, but the cost of their selectivity is that many available CCS OAH observations are not available through those compilations. In addition, there are a suite of nearshore ocean acidification, hypoxia, and temperature focused data collection efforts that use a variety of sensors and sampling techniques and have not yet been standardized or integrated. A deliberate synthesis of OAH-relevant datasets with standardized formatting and quality control maximizes our ability to explore, map, and resolve coastal climate stress on sub-regional scales (Bushinsky et al., 2019;

Chan et al., 2019). By including both discrete and validated autonomous sensor observations across depths and targeting all carbonate system and OAH-relevant parameters, this synthesis can complement the strengths of tightly focused compilations

such as SOCAT (Bakker et al., 2016) and CODAP-NA (Jiang et al., 2021). Additionally, by applying uniform QC standards and formatting to data across the CCS, this compilation builds on the usability, reliability, and spatiotemporal scale of currently available public nearshore compilations (e.g., Ruhl et al., 2021).


Here, we present the Multistressor Observations of Coastal Hypoxia and Acidification (MOCHA) synthesis, the highest resolution OAH-relevant U.S. West Coast dataset to date. MOCHA is a compilation of published nearshore temperature, dissolved oxygen, and carbonate chemistry-relevant datasets for the CCS and is newly archived and available at the National Centers for Environmental Information (NCEI, https://doi.org/10.25921/2vve-fh39; Kennedy et al., 2023) along with

associated metadata and quality assurance in adherence with the FAIR data management principles (Wilkinson et al., 2016). We source published data from within U.S. waters from oceanographic cruises, buoys, moorings, and shore samples as well as previously unpublished observations contributed by the author team, and present them in a formatted, quality-controlled, downloadable database for easy access and analysis by scientific teams across disciplines (Fig. 1). While this synthesis is not exhaustive, it highlights real disparities in oceanographic monitoring intensity and provides future investigators the

opportunity to compare and integrate their own datasets. This data compilation includes 13.7 million observations from 66 sources and spans from 1949-2020. To illustrate some of the synthesis product's potential uses, we further include and discuss several "case examples" that showcase the largest portion of the MOCHA dataset and its complementary strengths to SOCAT and CODAP-NA. However, we note that the compilation includes records at depth and those extending hundreds of kilometers offshore. It is our hope that this synthesis product supports scientific investigations at a wide range of spatial and

temporal scales and allows investigators to link between shallow and nearshore or coastal and oceanic environments. We anticipate that this synthesis product will be broadly useful to OAH-focused investigative teams and particularly impactful for coastal scientists investigating policy- and management-relevant projects, such as investigating spatiotemporal variation in marine climate risk from OAH events and warming, evaluating the efficacy and completeness of CCS monitoring efforts, linking oceanographic conditions to coastal social or socio-economic considerations across large geographic ranges (e.g.,

Ward et al., 2022), evaluating spatial management zones such as aquaculture sites (Clements and Chopin, 2016) and marine protected areas (e.g., Hamilton et al., 2003), and pursuing other questions of interest to coastal communities.

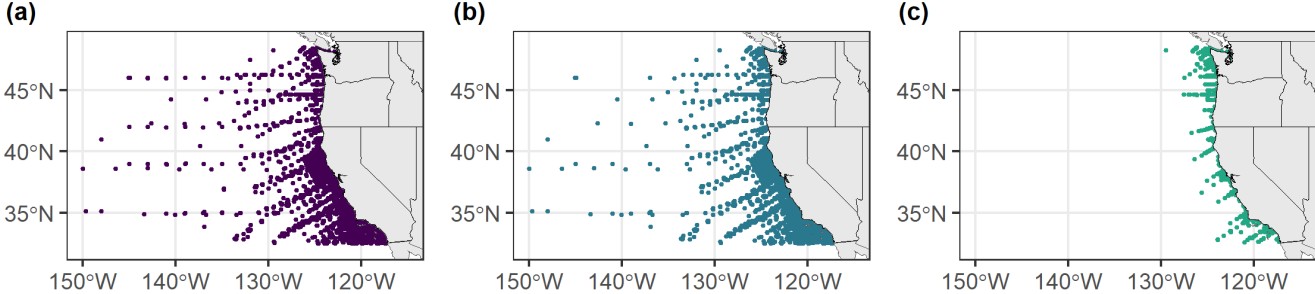

**Figure 1: All individual locations for temperature (a), dissolved oxygen (b), and pH (c) observations included in this synthesis along the U.S. West Coast. The pH extent fully captures the extent of all other carbonate system parameters. These figures overstate the useful spatial density of these data, as many individual locations have only been sampled once, but highlight the limited scale of available carbonate system observations relative to more commonly assessed parameters like temperature and dissolved oxygen.**

## 2. Methods

### 2.1 Data Sources and Types

This project compiled published and publicly available data, as well as previously unpublished data contributed by the author team, including multiparameter OAH-relevant observations from shipboard discrete water samples, *in-situ* autonomous sensors, and shore-collected samples from along the U.S. West Coast. We primarily sourced multiparameter data through existing public data portals, such as NCEI and the Ocean Observing Systems portals, but additionally contacted colleagues to request their assistance in locating additional datasets, presented the project at conferences and management meetings to collect community feedback on included datasets, and scanned published literature that likely included relevant datasets. We prioritized datasets that included carbonate system or dissolved oxygen observations in addition to temperature. When available alongside our target parameters, we also incorporated published chlorophyll and nutrient contents. In all cases, we took the published or publicly hosted data as our starting point, rather than asking for unprocessed data from the original investigators, then applied additional quality-control measures described in Sect. 2.4. We have limited this publication to data collected before 2020 and within U.S. waters, but we will continue to incorporate new observations according to the methods outlined below, where possible, and will periodically make updated versions of this synthesis dataset publicly available at NCEI (https://doi.org/10.25921/2vve-fh39; Kennedy et al., 2023) as support becomes available.

The data in this synthesis come from a wide array of observational methods and instruments. We screened carbonate system datasets before incorporating them following the discussions of method reliability summarized in Martz et al., (2015). The carbonate system observational methods adhere to one of the following observation methods: (1) discrete seawater samples, preserved at the time of collection and analyzed in a lab with established standards and techniques (e.g., Dickson and Sabine, 2010), of pH, total alkalinity (TA), and dissolved inorganic carbon (DIC); (2) pH measurements from ion-sensitive field-effect transistor-based autonomous sensors (e.g., Honeywell Durafet; Martz et al., 2010) or spectrophotometric sensors (e.g., SAMI-pH; Lai et al., 2018); and (3) $pCO_2$ measurements from autonomous equilibrium-based infrared gas analyzers (e.g., MAPCO2; Sutton et al., 2014) or spectrophotometric methods (e.g., SAMI-CO2; Schar et al., 2009). We did not include pH measured on glass electrode sensors, due to known issues with precision and calibration (Martz et al., 2010). We discarded any dissolved oxygen and carbonate system datasets that lacked accompanying temperature data. While we preferred carbonate system observations that also included salinity measurements, we retained pH and $pCO_2$ data without concurrent salinity measurements if they passed all other QC checks (e.g., Chan et al., 2017; Donham et al., 2023). Data collection methods are available for all parameters except temperature and salinity and have been simplified into four groups: 1)

"discrete", for bottle-collected samples analyzed in a laboratory, 2) "CTD" for observations from ship-side profiles with autonomous sensor arrays, 3) "autonomous sensors", for stationary instruments collecting data at pre-programmed intervals, and 4) "handheld sensors" for observations collected in the field via a glass-electrode probe. The specific instruments associated with each data source are available in the dataset metadata table in the Supplemental Information and archived at NCEI, Accession 0277984 (MOCHA_metadata_table_v2.csv; Kennedy et al., 2023).

## 2.2 Formatting

After identifying a dataset of interest, we downloaded all available processed data and metadata, including descriptive papers, primary investigator information, project and instrument descriptions, and the original source of the data. Each dataset was assigned a unique identifying number to ensure that every data point could be quickly associated with its parent data source and metadata (Table 1). For all datasets, we retained a copy of the original published data. We manipulated each original dataset into a comma-separated file with minimal alterations - typically limited to eliminating extra header rows and streamlining column names - before transferring datasets into R or Python for further formatting to ensure that all manipulations were trackable.

This synthesis dataset is structured such that each row represents an oceanographic observation from a shared time, depth, location, and data source which may include one or more individual parameter measurements. Parameter measurements are grouped with the parameter collection method, such as "discrete" or "autonomous sensor", and the data quality flag in adjacent columns. Additionally, all observations are also accompanied by "sample scheme" and "habitat" columns to facilitate easy data filtering. The sample scheme column classifies each dataset as one of four types: "cruise" for ship-collected samples, "mooring" for autonomous instruments attached to buoys, "intertidal/subtidal autonomous sensor" for shore- or diver-accessed autonomous sensors, and "intertidal/subtidal discrete collection" for water samples collected by hand from a dock or the shore. The habitat column identifies observations "estuarine" if they were collected within semi-restricted lagoons and bays (e.g., Humboldt Bay). All other observations are labeled as "oceanic". For a full description of included parameters, refer to the submission metadata archived at NCEI (SubmissionForm_carbon_v1_428.csv; Kennedy et al., 2023) and the dataset metadata table in the Supplemental Information.

We retained all directly measured chemical oceanographic observations as we incorporated each dataset, converted observations to standard units if necessary, and mapped them directly to our corresponding synthesis dataset columns. Fortuitously, all pH observations ingested into this compilation were already reported on the total pH scale. When necessary, we converted discrete pH observations reported at 25°C to *in-situ* conditions using accompanying temperature, salinity, pressure, carbonate-system, and nutrient contents using the R package seacarb (Gattuso et al., 2023). We used the following constants for these calculations: $K_1$ and $K_2$ from Lueker et al. (2000), $K_f$ from Perez and Fraga (1987), $K_s$ from Dickson (1990), and total boron concentrations from Uppstrom et al., (1974). We did not retain published data calculated from

algorithms or empirical relationships, such as TA calculated from a TA-salinity relationship or pH derived from temperature, salinity, and DO measurements (e.g., Alin et al., 2012). While we note that published data may have been summarized or filtered by the initial investigators, we did not further summarize or filter data before including it in this compilation except for the Ocean Observatories Initiative (OOI) moorings (dataset 66) discussed below.

| ID | Dataset | Primary location | Sampling scheme | Habitat | Parameters | Citation |
|---|---|---|---|---|---|---|
| 1 | Sea-surface water temperature, Santa Barbara Harbor | Santa Barbara LTER, CA | Intertidal/Subtidal discrete collection | Oceanic | T | Carter et al., 2021 |
| 2 | National Data Buoy Center Station BDXC1 | Bodega Head, CA | Mooring | Oceanic | T, S, Chl | National Data Buoy Center, 2023 |
| 3 | Mid-water SeaFET and $CO_2$ system chemistry at Alegria (ALE) | Santa Barbara LTER, CA | Mooring | Oceanic | T, S, pH, TA | Santa Barbara Coastal LTER et al., 2018 |
| 5 | West Coast Ocean Acidification Cruise 2016 | West Coast of the U.S. | Cruise | Oceanic | T, S, pH, DIC, TA, DO, Chl, Nuts | Alin et al., 2017 |
| 6 | National Data Buoy Center Station 46025 | Channel Islands, CA | Mooring | Oceanic | T, S | National Data Buoy Center, 2023 |
| 7 | National Data Buoy Center Station 46217 | Channel Islands, CA | Mooring | Oceanic | T | National Data Buoy Center, 2023 |
| 8 | National Data Buoy Center Station 46053 | Channel Islands, CA | Mooring | Oceanic | T, S | National Data Buoy Center, 2023 |
| 9 | National Data Buoy Center Station TDPC1 | Eureka, CA | Mooring | Oceanic | T, S, DO, Chl | National Data Buoy Center, 2023 |
| 10 | National Data Buoy Center Station FPXC1 | Fort Point, San Francisco Bay, CA | Mooring | Estuarine | T, S Chl | National Data Buoy Center, 2023 |
| 11 | National Data Buoy Center Station 46221 | Santa Monica Bay, CA | Mooring | Oceanic | T | National Data Buoy Center, 2023 |
| 12 | National Data Buoy Center Station 46235 | Imperial Beach, CA | Mooring | Oceanic | T | National Data Buoy Center, 2023 |
| 14 | National Data Buoy Center Station 46251 | Santa Cruz Basin, CA | Mooring | Oceanic | T | National Data Buoy Center, 2023 |
| 15 | National Data Buoy Center Station ICAC1 | Santa Monica, CA | Mooring | Oceanic | T | National Data Buoy Center, 2023 |
| 16 | National Data Buoy Center Station PRYC1 | Point Reyes, CA | Mooring | Oceanic | T | National Data Buoy Center, 2023 |
| 17 | National Data Buoy Center Station HBXC1 | Humboldt Bay, CA | Intertidal/Subtidal sensor deployment | Estuarine | T, S, DO, Chl | National Data Buoy Center, 2023 |
| 18 | National Data Buoy Center | Morro Bay, CA | Mooring | Estuarine | T, S, DO, Chl | National Data Buoy |

| | | | | | |
|---|---|---|---|---|---|
| | Station MBXC1 | | | | Center, 2023 |
| 19 | National Data Buoy Center Station MLSC1 | Moss Landing, CA | Mooring | Oceanic | T, S, DO | National Data Buoy Center, 2023 |
| 20 | National Data Buoy Center Station MTYC1 | Monterey, CA | Mooring | Oceanic | T, S, DO, Chl | National Data Buoy Center, 2023 |
| 21 | West Coast Ocean Acidification Cruise 2013 | West Coast of the U.S. | Cruise | Oceanic | T, S, pH, DIC, TA, DO, Chl, Nuts | Feely et al., 2015a |
| 22 | West Coast Ocean Acidification Cruise 2012 | West Coast of the U.S. | Cruise | Oceanic | T, S, DIC, TA, DO, Chl, Nuts | Feely et al., 2016b |
| 23 | West Coast Ocean Acidification Cruise 2011 | West Coast of the U.S. | Cruise | Oceanic | T, S, pH, DIC, TA, DO, Chl, Nuts | Feely et al., 2015b |
| 24 | West Coast Ocean Acidification Cruise 2007 | West Coast of the U.S. | Cruise | Oceanic | T, S, DIC, TA, DO, Nuts | Feely et al., 2013 |
| 25 | California Cooperative Oceanic Fisheries Investigations (CalCOFI) bottle database (1949 - 2019) | California | Cruise | Oceanic | T, S, DIC, TA, DO, Chl, Nuts | California Cooperative Oceanic Fisheries Investigations (CalCOFI), 2020 |
| 26 | Applied California Current Ecosystem Studies Partnership cruise observations (2013-2019) | Central California | Cruise | Oceanic | T, S, pH, TA, DO | Davis et al., 2018 *Previously unpublished data contributed by the authors. |
| 27 | UC Davis Coastal discrete ocean acidification dataset | West Coast | Intertidal/Subtidal discrete collection | Oceanic | T, S, pH, DIC, TA, DO | Feely et al., 2016a *Previously unpublished data contributed by the authors. |
| 28 | Bodega Marine Laboratory Weekly Horseshoe Cove discrete shore samples | Bodega Marine Laboratory, CA | Intertidal/Subtidal discrete collection | Oceanic | T, S, pH, DIC, TA, DO | *Previously unpublished data contributed by the authors. |
| 30 | Mid-water SeaFET pH and $CO_2$ system chemistry at Arroyo Quemado Reef (ARQ) | Santa Barbara LTER, CA | Mooring | Oceanic | T, S, pH, TA, DO | Santa Barbara Coastal LTER et al., 2020a |

| | | | | | | |
|---|---|---|---|---|---|---|
| 31 | Mid-water SeaFET pH and CO2 system chemistry with surface at Mohawk Reef (MKO) | Santa Barbara LTER, CA | Mooring | Oceanic | T, S, pH, TA, DO | Santa Barbara Coastal LTER et al., 2020b |
| 32 | Mid-water SeaFET pH and CO2 system chemistry at Santa Barbara Harbor/Stearns Wharf | Santa Barbara LTER, CA | Mooring | Oceanic | T, S, pH, TA, DO | Santa Barbara Coastal LTER et al., 2020c |
| 33 | Ocean Margin Ecosystems Group for Acidification Studies (OMEGAS) | West Coast | Intertidal/Subtidal sensor deployment | Oceanic | T, pH | Menge et al., 2015 |
| 34 | EAGER Project: pH/pCO2-sensing mooring platform on the Oregon coast | Oregon | Mooring | Oceanic | T, $pCO_2$ | Chan et al., 2012 |
| 35 | NH10 mooring SAMI-$CO_2$ time-series | Oregon | Mooring | Oceanic | T, S, pH, $pCO_2$ | DeGrandpre, 2016 |
| 36 | SB LTER calibration water-sample pH and CO2 system chemistry | Santa Barbara LTER, CA | Cruise | Oceanic | T, S, pH, DIC, TA | Santa Barbara Coastal LTER et al., 2022 |
| 37 | Bodega Marine Reserve monthly shore samples | Bodega Marine Reserve, CA | Intertidal/Subtidal discrete collection | Oceanic | T, S, pH, DIC, TA, DO | *Previously unpublished data contributed by the authors. |
| 39 | California Coastal seagrass project | California | Intertidal/Subtidal sensor deployment | Varies by site | T, S, pH, TA, DO | Ricart et al., 2021 |
| 40 | California kelp forest tidal FET sites | California | Intertidal/Subtidal sensor deployment | Oceanic | T, pH, DO | Kroeker et al., 2023 |
| 41 | NOAA Northwest Pacific harmful algal bloom program cruise SH1709 | Washington and Oregon | Cruise | Oceanic | T, S, DIC, TA, DO, Nuts | Alin et al., 2019 |

| 42 | Oceanographic cruise calibration and validation samples of California Current Ecosystem | Southern California Bight | Cruise | Oceanic | T, S, DIC, TA, DO, Chl, Nuts | Send et al., 2016 |
|----|------|------|------|------|------|------|
| 43 | CCE1 mooring $pCO_2$ time-series | Point Conception, CA | Mooring | Oceanic | T, S, pH, $pCO_2$, $fCO_2$, DO | Sutton et al., 2016b |
| 44 | CCE2 mooring $pCO_2$ time series | Point Conception, CA | Mooring | Oceanic | T, S, pH, $pCO_2$, $fCO_2$, DO | Sutton et al., 2012 |
| 45 | CeNCOOS in situ water monitoring data at Trinidad Head, California | Trinidad, CA | Intertidal/Subtidal sensor deployment | Oceanic | T, S, DO, Chl | Shaughnessy, 2023 |
| 46 | SFSU Estuary and Ocean Science Department YSI | Tiburon Peninsula, CA | Intertidal/Subtidal sensor | Estuarine | T, S, Chl | Dewitt, 2022 |
| 47 | CeNCOOS water monitoring data at the Santa Cruz municipal wharf | Santa Cruz, CA | Intertidal/Subtidal sensor deployment | Oceanic | T, S, DO, Chl | Kudela, 2020 |
| 49 | San Francisco Estuary Institute and the Aquatic Science Center Regional Monitoring Program | San Francisco Bay, CA | Cruise | Estuarine | T, S, DO, CHl | Bezalel et al., 2021 |
| 50 | West Coast Estuary Data: Santa Monica Bay | Santa Monica | Mooring | Oceanic | T, S, pH, $pCO_2$, DO | Rosenau et al., 2021a |
| 51 | West Coast Estuary Data: San Francisco Bay | SF Bay | Mooring | Estuarine | T, S, pH, DO, Chl | Rosenau et al., 2021a |
| 52 | Validation discrete observations for the Cha Ba mooring | La Push, WA | Cruise | Oceanic | T, S, DIC, TA, Nuts | Alin et al., 2016 |
| 53 | Morro Bay BM1 T-Pier (NOAA Station MBXC1) | Morro Bay, CA | Mooring | Estuarine | T, S, pH, DO, Chl | Walter, 2023 |
| 54 | Morro Bay BS1 Station | Morro Bay, CA | Mooring | Estuarine | T, S, pH, DO, Chl | California Polytechnic State University, 2023 |

| 55 | Cape Elizabeth mooring MAPCO2 time-series | Cape Elizabeth, WA | Mooring | Oceanic | T, S, pH, pCO$_2$, fCO$_2$, DO | Sutton et al., 2013 |
|----|-------------------------------------------|--------------------|---------|---------|-------------------------------|---------------------|
| 56 | Stillwater Cove TidalFET | Carmel, CA | Intertidal/Subtidal sensor deployment | Oceanic | T, S, pH, DO | Donham, 2022a |
| 57 | National Data Buoy Center Station 46211 | Grays Harbor, WA | Mooring | Oceanic | T | National Data Buoy Center, 2023 |
| 58 | National Data Buoy Center Station NEAW1 | Neah Bay, WA | Mooring | Estuarine | T | National Data Buoy Center, 2023 |
| 59 | National Data Buoy Center Station CECC1 – 9419750 | Crescent City, CA | Intertidal/Subtidal sensor deployment | Oceanic | T | National Data Buoy Center, 2023 |
| 60 | National Data Buoy Center Station 46237 | San Francisco, CA | Mooring | Oceanic | T | National Data Buoy Center, 2023 |
| 61 | National Data Buoy Center Station 46240 | Monterey Bay, CA | Mooring | Oceanic | T | National Data Buoy Center, 2023 |
| 62 | National Data Buoy Center Station PORO3 | Port Orford, OR | Mooring | Oceanic | T | National Data Buoy Center, 2023 |
| 63 | National Data Buoy Center Station CHAO3 | Charleston, OR | Mooring | Estuarine | T | National Data Buoy Center, 2023 |
| 64 | CB-06 mooring MAPCO2 time-series | Coos Bay, OR | Mooring | Oceanic | T, S, pH, pCO$_2$, fCO$_2$, DO, Chl | Sutton et al., 2019 |
| 65 | NH10 mooring MAPCO2 time-series | Newport, OR | Mooring | Oceanic | T, S, pH, pCO$_2$, fCO$_2$, DO, Chl | Sutton et al., 2016a |
| 66 | Ocean Observatories Initiative (OOI) Washington and Oregon inshore and shelf moorings | Washington and Oregon | Mooring | Oceanic | T, pH, DO | NSF Ocean Observatories Initiative, 2022 |
| 67 | Trinidad Head Line CTD Hydrography | Northern California | Cruise | Oceanic | T, S, pH, DO | Bjorkstedt, 2023 |
| 68 | Newport Hydrographic Line CTD casts 1997–2021 | Central Oregon | Cruise | Oceanic | T, S, DO | Risien et al., 2022b |

| 69 | Oregon's Marine Reserve mooring | Oregon | Mooring | Oceanic | T, DO | Aylesworth et al., 2022 |
|---|---|---|---|---|---|---|
| 70 | CMOP Saturn-02 mooring | Columbia River Estuary, OR | Mooring | Estuarine | T, S, DO | Columbia River Intertribal Fish Commission Center for Coastal Margin Observation and Prediction, 2023 |
| 71 | Monthly cross-shore transects of biogeochemical properties in La Jolla, CA | Southern CA | Cruise | Oceanic | T, S, pH, DIC, TA, DO, Nuts | Kekuewa and Andersson, 2022 |

**Table 1: Overview of the included data sources in the MOCHA compilation. Potential measured parameters for each dataset include temperature (T), salinity (S), pH, partial pressure of $CO_2$ ($pCO_2$), fugacity of $CO_2$ ($fCO_2$), dissolved inorganic carbon (DIC), total alkalinity (TA), dissolved oxygen (DO), chlorophyll-A (Chl), and nutrients (Nuts). Users need to be mindful of the difference between climate-quality and weather-quality datasets and assess the suitability of these datasets for their needs (Newton et al., 2015). The origins of all the included datasets in this compilation are further described in the dataset metadata table available in the paper Supplement and archived at NCEI (https://doi.org/10.25921/2vve-fh39, dataset_metadata_table_v2.csv, Kennedy et al., 2023). Additional, detailed discussions of the following datasets have been previously published: 5 (Feely et al., 2008); 21-24, 26 (Feely et al., 2016a); 25 (Bograd et al., 2003); 26 (Davis et al., 2018); 33 (Chan et al., 2017); 49 (Salop and Herrmann, 2019); 50 and 51 (Rosenau et al., 2021b); 56 (Donham et al., 2022b); 66 (Trowbridge et al., 2019); 67 (Bjorkstedt and Peterson, 2015); 68 (Risien et al., 2022a); 69 (Barth et al., 2021); 70 (Baptista et al., 2015); and 71 (Kekuewa et al., 2022).**

## 2.3 Dataset 66: Ocean Observatories Initiative (OOI) Moorings

The Washington and Oregon OOI mooring data (dataset 66) included millions of observations of temperature, salinity, dissolved oxygen, pH, and $pCO_2$ at sub-minute resolutions. The size of these datasets required us to aggregate the data to daily mean values before incorporation into the larger synthesis dataset. Because many of these OOI data had not been previously quality controlled, we contacted OOI staff for their guidance on initially filtering the raw data before aggregation. They provided extensive code developed by the sensor manufacturers and OOI staff to identify erroneous pH and DO data from the raw publicly available streams, available at https://github.com/oceanobservatories/ooi-data-explorations/tree/master/python, as well as significant protocol guidance that has since been made public (Palevsky et al., 2022). OOI staff also provided access to discrete sample analyses taken at the sensor moorings to further ground-truth measurements. We only retained data for aggregation if it 1) passed through the provided OOI code's automated checks, 2) had discrete samples associated with the beginning and end of that sensor's deployment, 3) the daily mean sensor values for DO and pH on the day of discrete sampling were within 20 μmol kg$^{-1}$ of the discrete sample dissolved oxygen and/or 0.05 pH units, and 4) displayed reasonable DO content and pH values and variance over time, following OOI's suggested protocols for both automated and "human in the loop" quality control practices (Palevsky et al., 2022). We eliminated all DO data collected prior to 2018 based on advice of OOI staff because the DO sensors prior did not have adequate biofouling control. We then aggregated these data into daily mean values before formatting and quality controlling them further following the practices described for all other incorporated datasets and described in Sect 2.4.

## 2.4 Quality Control

After formatting individual datasets, we checked all observations to standardize quality across datasets and to avoid using questionable data in future analyses. This quality standardization did not extend to raising all datasets to a "climate-quality" standard (Newton et al., 2015). Users of these data should be aware of the difference between climate-quality versus weather-quality data, as both types of data are included in this synthesis and often coexist within the same datasets. Our quality assurance/quality control (QA/QC) methods drew from a combination of the publishing authors' notes, plots of the data, and expert knowledge of the CCS. The majority of our incorporated datasets had been previously published and subjected to at least automated QA/QC processes, but additional "human in the loop" secondary QC was necessary for almost all datasets, particularly those from autonomous sensors (Pavlevsky et al., 2022). Incoming quality-control notes associated with each data source ranged widely, though most datasets that did include quality information followed the Quality Assurance/Quality Control of Real-Time Oceanographic Data (QARTOD) system, which assigns flags based on internal instrument checks, data reasonableness, and collection method (Bushnell 2018). Given the variability in flagging schemes that incorporated datasets used and the impossibility of accurately assigning detailed QARTOD-style flags to datasets that did not include similarly detailed notes, we opted to create a simpler, three-level quality scheme that could be applied to all datasets. Using available existing QA/QC information and our further quality control investigations, we categorized each individual parameter measurement as one of three confidence levels: 1 for "plausible and reasonable" data, 2 for data that we had not assessed, and 3 for "low quality or unreliable" data. We flagged all data the publishing authors had listed as unreliable or suspect with a 3. Regardless of published notes, we assigned all other observations a flag of 2 before further evaluation by our team.

Given the diversity of the datasets and projects this synthesis draws from, we examined each dataset individually using a combination of plots tailored to maximize our ability to identify and evaluate anomalies in that dataset's specific oceanographic and spatiotemporal context. Given that this synthesis primarily sourced published data, we erred towards retaining data as "plausible", rather than following a more stringent flagging philosophy. We recommend that investigators perform additional QC with the MOCHA dataset targeted towards their project requirements. Common quality control plotting techniques included property-property plots of temperature, salinity, DO, pH, TA, and DIC against one another; single-parameter time series from sensor and long-running datasets; and map views and oceanographic cross sections of synoptic cruise data. We examined questionable data through as many different views as possible, such as examining apparent outliers in a temperature-salinity property-property plot individually in their respective time series, to ensure that we were not flagging real or plausible observations. When possible, we further evaluated suspicious observations against other datasets collected nearby. We discussed all data flagging decisions with at least three project members. After this focused quality control, all observations not flagged as "low quality or unreliable" (3) were upgraded to our "plausible and reliable" flag (1) with the exception of 300 surf zone DO measurements taken from shore, which were left as "unevaluated"

(2) since they do not reflect oceanic conditions. All subsequent mapping and analysis with the observed oceanographic values used only "plausible and reliable" data. For a full example of our formatting and flagging practices, please refer to the Supplemental Information.

## 2.5 Example Subset: Daily Data

High-resolution (sub-daily observations) autonomous sensors are an important component of this synthesis dataset, but the data they produce come with significant computational costs. Furthermore, variability on the scales of hours or minutes captured by such high-resolution records is less comparable to lower-resolution datasets such as those collected over quarterly or annual synoptic oceanographic cruises. To evaluate the spatiotemporal extent of our data coverage, seasonal patterns, and relationships between observed parameters, we created an aggregated summary dataset of daily mean values for each location, depth, and data source. We dropped all questionable individual parameter measurements (i.e., data flagged with a "3" QA/QC code) before creating this summary dataset to ensure that unreliable data did not influence averages. The daily averaging reduced the number of observations (rows) from 13.7 million to 1.2 million as high-resolution sensor datasets, some with observations every 20 minutes, were collapsed into a single row per day. We used this summary dataset in all following example cases that do not explicitly cite "original data." This aggregated summary dataset is available alongside the full MOCHA compilation at NCEI (aggregated_daily_dataset.csv, Kennedy et al., 2023) and we have included the code necessary to recreate it in our public code repository (https://github.com/egkennedy/DSP_public_code).

## 3 Results and Discussion

### 3.1 Overall Data Totals

This synthesis dataset includes observations from 66 individual data sources organized across 13.7 million rows ("observations") and 41 columns and spans from 1949-2020. This includes 24.1 million individual parameter measurements, with 13.2 million temperature, 3.6 million salinity, 3.3 million DO, 2.1 million pH, 1.2 million chlorophyll, 561,000 nutrient, 113,000 $pCO_2$, 9,300 TA, and 8,300 DIC measurements. While we prioritized multiparameter datasets for this effort, our synthesis also includes several temperature-only, high-resolution records to fill specific project needs. The full suite of carbonate system parameters can be directly calculated from 48,000 observations with two reliable carbonate system parameter observations and co-occurring reliable temperature and salinity measurements.

Across sampling schemes, moorings contribute the bulk of the MOCHA observations with 8.9 million rows, followed by intertidal or subtidal autonomous sensors with 3.8 million, oceanographic cruise observations (which include CTD profiles) with 98,000, and finally intertidal and subtidal discrete collections with 24,000. By measurement method, autonomous sensors are the most common, contributing 5 million individual measurements, versus 224,000 individual discrete measurements, 193,000 CTD measurements, and 828 handheld field sensor measurements.

 **3.2 Aggregated Daily Data Totals**

Summarizing the data by day for each dataset, location, and depth provides a clearer picture of the availability of multiparameter data by diminishing the outsized influence of high-resolution temperature sensors. Of the 1.2 million daily averaged observations, just 104,000 are temperature-only. Individual parameter totals are shown in Table 2. Full carbonate system calculations could be performed on 12,000 of the daily observations with measurements of temperature, salinity, and 295 two of the principal carbonate system parameters. As with the disaggregated, full dataset, data totals varied substantially by measurement method and autonomous sensors are still the most common, contributing 643,000 individual daily averaged parameter measurements versus 223,000 discrete, 192,000 CTD, and 816 handheld sensor measurements.

| Parameter | Collection Method | Daily Total Observations | Overall Reliability Rate |
|---|---|---|---|
| DO | discrete | 199,816 | 99.7% |
| | autonomous sensor | 563,885 | 92.4% |
| | CTD | 128,562 | 99.9% |
| | handheld sensor | 382 | 93.2% |
| pH | discrete | 4,068 | 99.6% |
| | autonomous sensor | 78,894 | 88.7% |
| | CTD | 63,404 | 100% |
| DIC | discrete | 8,211 | 99.1% |
| TA | discrete | 8,858 | 98.2% |

**Table 2: Overview of dissolved oxygen (DO), pH, dissolved inorganic carbon (DIC), and total alkalinity (TA) observation methods,**
**number of daily observations (grouped by data source, location, and depth), and the overall reliability rates. Autonomous sensors are associated with slightly lower reliability rates due to periods of sensor biofouling or malfunction.**

**3.3 Flagging and Reliability**

The amount of original data flagged as unreliable varied substantially by dataset, parameter, and observation method, but was typically low (Fig. 2). As the bulk of the data in this synthesis product were previously published and had undergone 305 some preliminary QA/QC prior to our incorporation, high reliability rates were expected. Of the dozens of datasets contributing temperature and salinity observations, only one dataset had a parameter flag rate above 5%. Flag rates above 10% were uncommon for all parameters across all datasets, and completely absent for TA and DIC observations. For pH and DO, flag rates within datasets were above 10% for three and eight datasets, respectively. In each case, high rates of

"unreliable" data were caused by (1) clear periods of autonomous sensor malfunction, (2) observational methods described
by the publishing authors as unreliable, or (3) more rarely, intentionally higher QA/QC standards applied to data which had
not been previously screened and published. The vulnerability of autonomous sensors to periods of biofouling or sensor
malfunction contributed to higher flag rates relative to other methods, but all four methods were largely reliable (Table 2).
Across the entire MOCHA compilation, 99.8% of temperature, 96.8% of salinity, 93.1% of DO, 89.1% of pH, 99.1% of
DIC, and 98.2% of TA measurements were considered "reliable or plausible". Across all individual measurements, 97.3%
are classified as reliable.

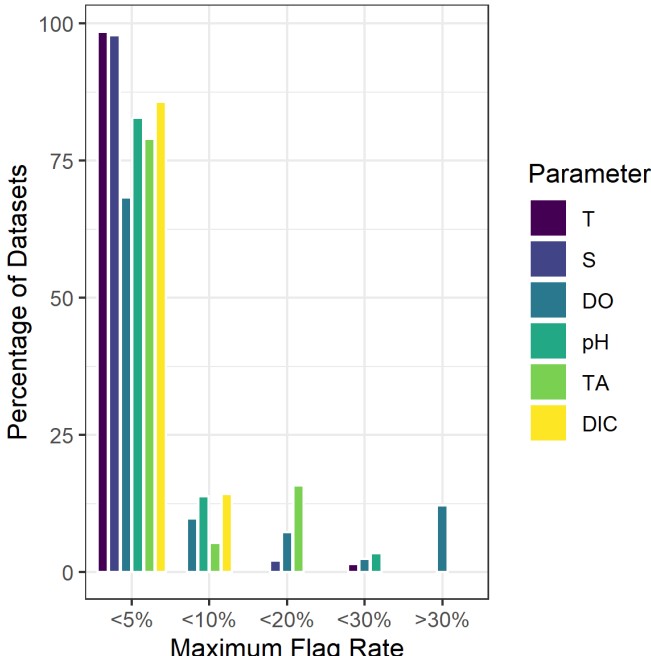

**Figure 2: The rate of unreliable ("flagged") observations varied by dataset and parameter measured between temperature (T), salinity (S), dissolved oxygen (DO), pH, total alkalinity (TA), and dissolved inorganic carbon (DIC). Maximum unreliable flag rates were generally low, especially for T and S. All datasets that included measurements with > 30% flag rates used measurement methods described by the original publishers as "not quantitative". Flag rates between 10% and 30% were uncommon but reflected occasional periods of fouling or equipment malfunction in high resolution autonomous sensor datasets or, in rare cases, more stringent standards applied to datasets that had not been previously published and initially quality controlled.**

### 3.4 Spatiotemporal Data Distribution

This dataset spans the U.S. West Coast and reflects the spatiotemporal bias of observational records. Observations are more
common in nearshore, near-surface environments and exhibit greater sampling effort in recent years. Fifty-six percent of
daily observations were collected within 50 km of shore and 27% within 25 m of the surface. Eighty-six percent of all daily
observations were collected after 1990. Carbonate system observations are especially skewed toward recent years, with no
measurements of pH, TA, DIC, or $pCO_2$ in this compilation prior to 2006. By contrast, temperature, salinity, and DO records
are common after 1980.

The spatiotemporal coverage of our dataset is highly variable, though generally improves through time. Mapping the density of observations within 50 km of the coastline and 25 m of the surface through time highlights the influence of dense coastal human populations and major research institutions (Fig. 3). By contrast, the region between 38° N and 44° N is much less densely observed and loses considerable oceanographic monitoring capacity between 2015 and 2020. Temperature and DO measurements have the most extensive coverage but are sparse outside of Southern California before 2000. Salinity measurement density hews closely to the DO distribution and, as such, is not shown here. After 2015, carbonate system observations are limited to a few locations with sporadic coverage north of 39° N which correspond to pH and $pCO_2$ moorings. Overall, this data compilation demonstrates large spatial and temporal data gaps, which limit our ability to resolve rapid changes in ocean acidification, hypoxia, or warming risk or to contextualize current oceanographic conditions with respect to the recent past.

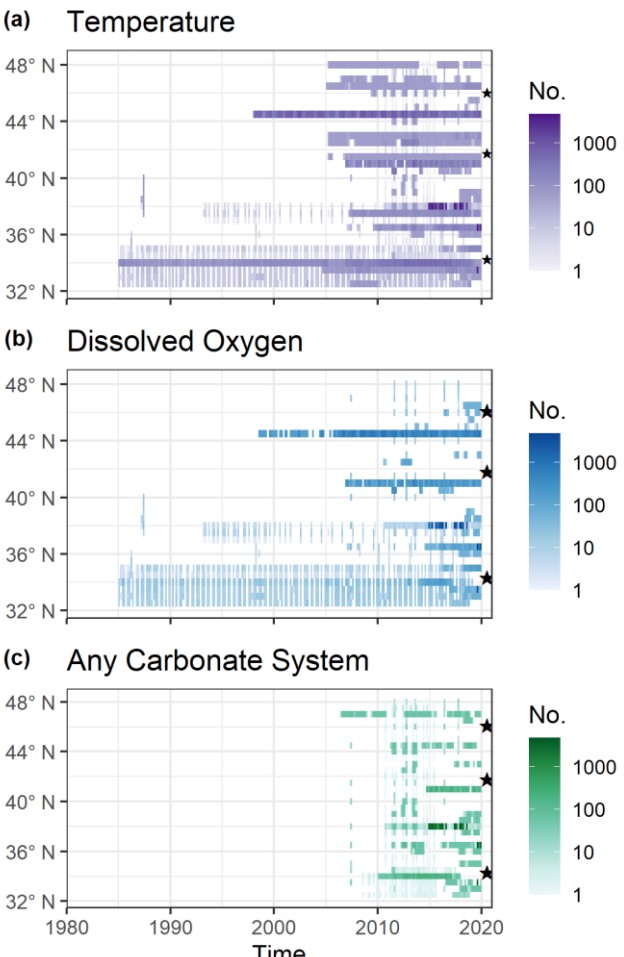

**Figure 3: The number of measurements within 50 km of the shore and 25 m of the surface for temperature (a), dissolved oxygen (b), and any carbonate system measurements (c) using two-month, 0.5-degree latitude spatiotemporal blocks. Salinity (not shown)**

**hews closely to the dissolved oxygen distribution. From north to south, stars mark the Washington-Oregon border, the Oregon-**
**California border, and Point Conception (34.5º N). Spatial data coverage was best across all parameters between 2010 and 2015, whereas overall observation quantity was highest between 2015 and 2020. Since 2015, dissolved oxygen and carbonate system measurements have become more concentrated into fewer locations along the coast despite increasing awareness of the risks of nearshore acidification and hypoxia events.**

The intra-annual distribution of the daily data is more complex than the interannual distribution (Fig. 4). Temperature, salinity, and DO records are common throughout the year, but have distinct peaks in abundance in April, May, and July through September. Carbonate system records are patchier temporally. Nearly 50% of all TA and DIC observations were taken in May or August, with an additional 19% of observations from September, reflecting the sampling months of the NOAA West Coast Ocean Acidification cruises (Feely et al., 2016a). Between October and April, no single month includes

more than 8% of DIC observations or 5% of TA observations. pH observations are more evenly distributed throughout the year, with each month hosting 6-10.5% of the observations except August, which hosts 16%. The concentration of carbonate system observations between May and September is particularly concerning, as upwelling season in Central and Southern California starts in earnest in April (García-Reyes and Largier, 2012; Jacox et al., 2018) at least two principal carbonate system parameters must be measured to fully constrain the carbonate system (Dickson and Sabine, 2010), so the

observational record may be missing significant low pH, low DO events from the early upwelling season.

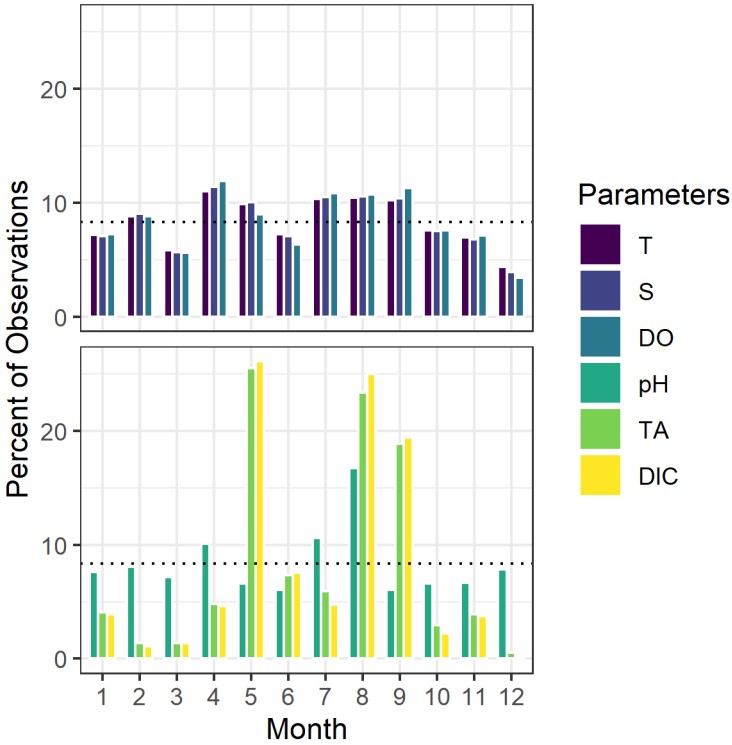

**Figure 4: The distribution of daily observations by month varies substantially by parameter relative to an equal split (dashed lines). Temperature (T), salinity (S), and dissolved oxygen (DO) observations are fairly evenly distributed across seasons, with**

notable observational peaks in April, May, July, August, and September. Carbonate system parameters (pH, total alkalinity or TA, and dissolved inorganic carbon or DIC) are more concentrated in the summer months, with nearly all TA and DIC observations occurring in May, August, or September. Of the carbonate system parameters, only pH observations are nearly equitably distributed throughout the year.

## 3.6 Oceanographic Analysis Case Examples

### 3.6.1 Monthly Climatology

This synthesis dataset supports several avenues of investigation of the relationships between OAH parameters. For example, evaluating the variations in monthly climatology across OAH parameters in waters shoreward of the 100 m depth contour shows intriguing differences between regions (Fig. 5). Temperatures rise in all regions during the spring and summer months, peaking between July and September. In Washington and Oregon, peak upwelling occurs between June and August (Bograd et al., 2009; Jacox et al., 2016), which coincides with the period of highest variability and lowest minima for pH and DO observations captured in this synthesis. In both California regions, separated at Point Conception (34.5° N), seasonal surface data are less consistent with the expected upwelling patterns. There, peak upwelling occurs between April and June and is weakest in Southern California (Bograd et al., 2009; García-Reyes and Largier, 2012; Jacox et al., 2016). Somewhat unexpectedly, the lowest median DO and pH observations occur between July and September in both California regions rather than during the months of expected peak upwelling. This trend may reflect intermittent upwelling into the warmer summer months or could be capturing high surface respiration as waters warm; conclusive evidence of either phenomenon requires further investigation. October through March conditions across all West Coast regions are more sparsely sampled, but have less variability, cooler mean temperatures, and higher dissolved oxygen content and pH.

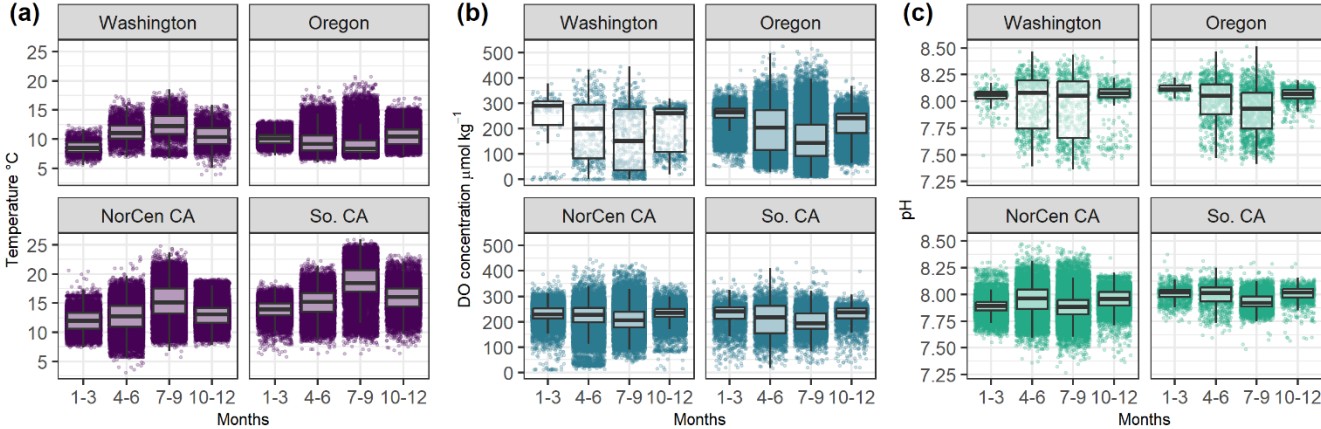

Figure 5: Measurements shoreward of the 100 m bathymetric contour of temperature (a), dissolved oxygen (DO) (b), and pH (c) capture intra-annual and regional variation. The lowest median DO and pH conditions are found with the highest temperatures in late summer, rather than during peak upwelling periods (April – June). Here, California is split into two regions: NorCen CA, spanning the northern border to Point Conception (34.5 N), and So. CA, from Point Conception to the southern border. Ninety-nine percent of the data fall within 30 km of shore and 65% falls within 10 km of shore.

 **3.6.2 Shallow OAH Events**

Nearshore OAH vulnerability information can be particularly important for effective coastal management (Ekstrom et al., 2015; Woodson et al., 2018). Within state waters (< 5 km from shore) in the surface 50 m, there are thousands of co-occurring observations of pH below 7.8 and DO below commonly applied hypoxia thresholds (Fig. 6; e.g., Vaquer-Sunyer and Duarte, 2007; Hoffman et al., 2011). pH conditions below 7.8 can be stressful for many marine organisms (e.g., Byrne and Przeslawski, 2013; Gobler and Baumann, 2016; Bednaršek et al., 2021; Kroeker et al, 2023) and have been observed 8,665 times within 5 km of shore and 50 m of the surface in this data compilation. Of these instances, 65 observations are accompanied by DO contents below the "coastal hypoxia" threshold of 61 $\mu$mol kg$^{-1}$ and 400 observations have DO contents below the "mild hypoxia" threshold of 107 $\mu$mol kg$^{-1}$ (Hofmann et al., 2011). An additional 220 of these near-surface observations of DO contents below 61 $\mu$mol kg$^{-1}$ in state waters have been recorded without accompanying pH information. No simultaneous surface observations of DO and pH record coastal hypoxic conditions with pH levels above 7.8. The low pH, low oxygen observations are most common off the Oregon coast and are typically associated with low temperature upwelling events, but simultaneous mild to moderately hypoxic and low pH conditions are also found occasionally throughout the coast and at a range of temperatures, especially during late summer in semi-restricted estuaries. The few simultaneous observations of DO content and pH suggest that fewer than 1% of observations of low pH (pH < 7.8) in state waters are accompanied by hypoxic water, whereas shallow hypoxic state waters might always be accompanied by low pH conditions. These relationships underscore the importance of multiparameter OAH observations, the clear need for pH monitoring efforts to catch up with DO monitoring efforts, and the potential for even shallow waters to experience extreme conditions.

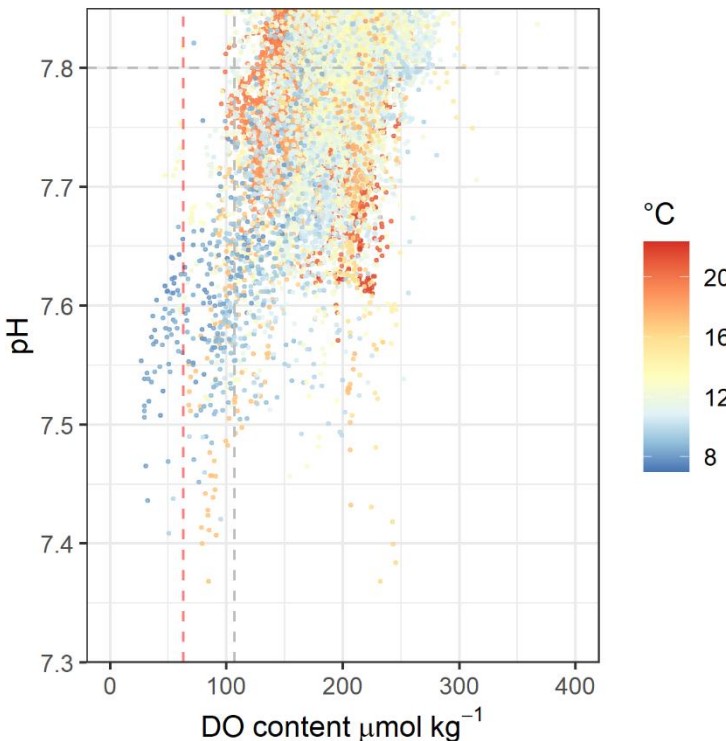

**Figure 6: Low dissolved oxygen (DO) and pH conditions are frequently present in state waters (within 5 km of the shore) and 50 m of the surface. pH measurements below 7.8 (grey dashed horizontal line) are common but are more rarely accompanied by mildly hypoxic (< 107 μmol kg⁻¹ or 3.5 mg L⁻¹ DO, grey dashed vertical line) or hypoxic (< 61 μmol kg⁻¹ or 2 mg L⁻¹ DO, red dashed vertical line) conditions. Simultaneous low pH, low DO events are typically associated with low temperatures, whereas low pH conditions alone are present across a wide range of temperatures.**

### 415 3.6.3 Total Alkalinity-Salinity Relationships

As a final example usage, we used the MOCHA synthesis to explore surface (< 25 m depth) TA-salinity relationships along the coast. Developing robust TA-salinity relationships for near-surface, nearshore waters has produced intense interest. Because salinity observations are more readily available in the historical record and relatively cheap to reliably collect, robust TA-salinity relationship or algorithms allow the full carbonate system to be calculated while only directly measuring

one principal parameter; however, these relationships and algorithms can be hampered by nearshore variability (e.g., Fassbender et al., 2017, Davis et al., 2018). We examined surface (< 25 m depth) discrete TA and salinity observations from within 100 km of the shore along the Washington, Oregon, and California coasts and compared the data collected within 2 km of shore to those collected between 2 and 100 km from shore (Fig. 7). Our TA-salinity relationships were very similar when using a 50 km and 100 km cutoff distance and we show the more extensive data here for closer comparisons with

previous investigators. Our TA-salinity slopes were not significantly different between any Washington and Oregon regions, though we note that our in our compilation, Washington and Oregon both have very limited discrete TA data within 2 km of shore, which produced large standard errors in the slope terms (4.5 and 3 μmol kg⁻¹, respectively). Our observed offshore

Washington TA-salinity relationship of $TA = 42.2 \pm 1.2 \times S + 823$ is more comparable to the Wootton and Pfister (2012) regression, which centered off the Strait of Juan de Fuca, than that from Fassbender et al. (2017). However, we did not correct for seasonal or watershed biases in this example and focus on a more limited stretch of nearshore waters, which may account the differences between our calculated relationships and that of Fassbender et al. (2017).

Each of the two California regions, split at Point Conception (34.5⁰ N), have TA-salinity regressions that are statistically distinct from each other and from both Pacific Northwest regions. The offshore California slope terms are much larger than in the Pacific Northwest region and significantly larger than the Cullison Gray et al. (2011) salinity coefficient of 50.8, particularly our slope for the Northern and Central region ($57.4 \pm 0.9$ μmol kg$^{-1}$). At a salinity of 33.5, these differences produce an increase in estimated TA of 94.3 μmol kg$^{-1}$ between our calculated Northern and Central California relationship and the Cullison Gray et al. (2011) relationship, which translates to an increase in estimated aragonite saturation of 0.1 at 12°C and pH = 8.1. The Cullison Gray et al. (2011) relationship was derived from unpublished $pCO_2$ and DIC observations, all taken prior to 2007, so there is limited temporal overlap between our sample sets and any spatial differences in sample area cannot be assessed. The California nearshore region is well-sampled relative to the Pacific Northwest and displays significant variability, potentially reflecting local differences in bedrock or organic alkalinity contributions. The variability in nearshore TA-salinity relationships will continue to present a challenge for coastal communities and state agencies, underscoring the importance of monitoring multiple parameters of the carbonate system in highly nearshore environments.

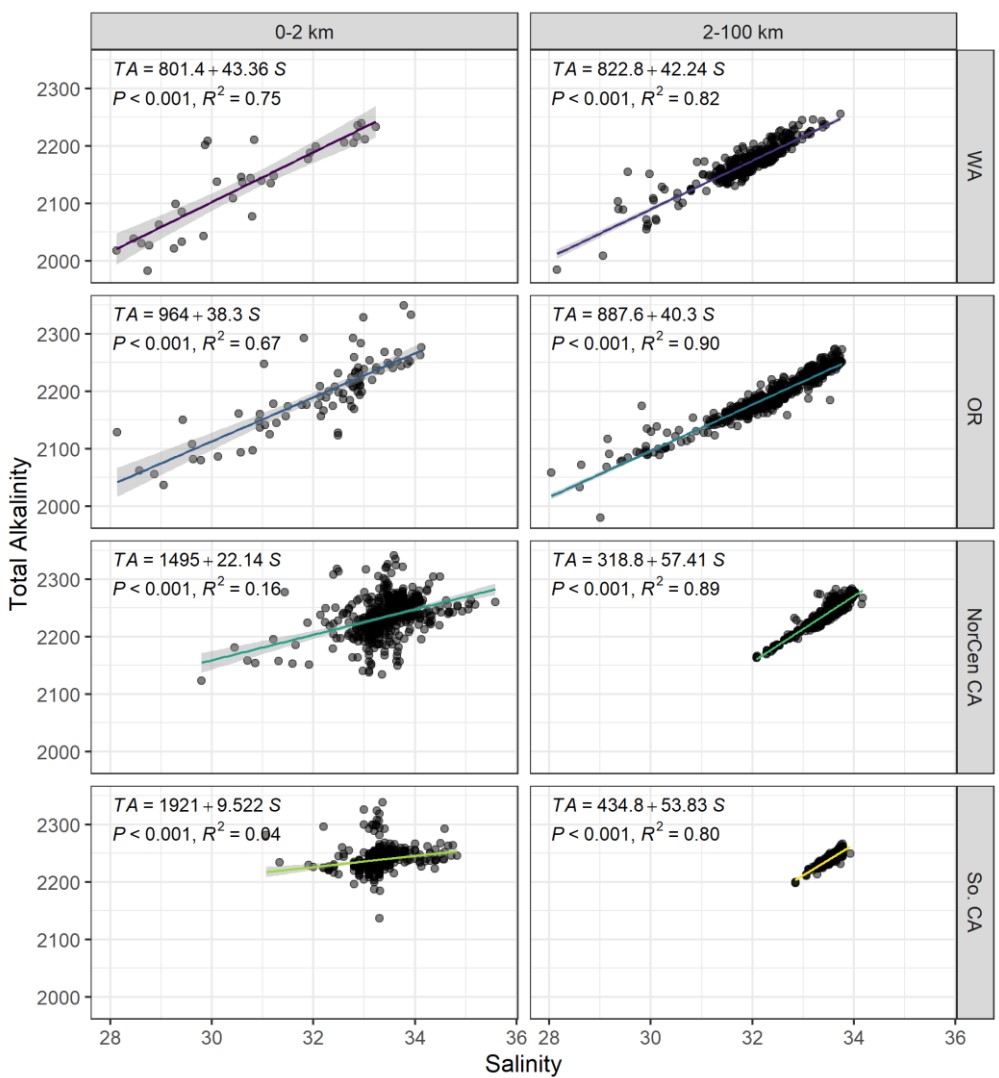

**Figure 7: Regional near-surface (< 25 m) total alkalinity (TA)-salinity relationships from 0-2 km from shore and 2-100 km offshore in along the U.S. West Coast. As with Figure 5, the break between northern and southern CA is Point Conception, at 34.5° N. These relationships reflect only direct measurements of salinity and TA on discrete samples with salinity > 28.**

### 3.7 Dataset Limitations

This data compilation reflects reliable, publicly available data, and directly contributes to our ability to map coastal temperature, DO, and carbonate system variation; however, this synthesis also encodes the limitations of the current observational record and the differences in data availability, data scales, and data quality. High resolution autonomous sensors provide excellent temporal resolution for a specific location, but are vulnerable to sensor drift, are not often published with clear calibration records, and are rarely deployed in arrays that fully capture the carbonate system as well as

temperature and DO variability. Conversely, discrete samples and CTD profiles from synoptic cruises provide extremely

high-precision, multiparameter observations with broad spatial resolution, but are less relatable to high-resolution sensors or hand-collected observations from the surf zone. Carbonate system observation availability has a strong seasonal and spatial bias, with data concentrated in summer months and along coastal population centers. The MOCHA synthesis pulls these distinct data sources into a single synthesis product, but we do not claim to have fully resolved the inherent difficulties of combining data of differing quantity, resolution, and quality into a unified picture of the nearshore CCS.

Additional data streams that provide both spatial and temporal resolution could help bridge some of the divides between quality, quantity, and spatial extent in this synthesis and we acknowledge a few such potential data streams here. The temperature and dissolved oxygen records do not include CTD casts from most annual fishery-independent surveys, which could improve spatial resolution at all depths (e.g., Sakuma, 2022). This compilation also excludes some valuable carbonate system data streams, particularly those focused on $pCO_2$ measurements currently available through SOCAT (Sabine et al., 2013; Bakker et al., 2016). Additional potential carbonate system data sources include pH or $pCO_2$ records from autonomous gliders (e.g., Chavez et al., 2017) and $pCO_2$ and DIC records from shore-based monitoring systems (e.g., Burke-o-Lators; Hales et al., 2004; Bandstra et al., 2006). The first would significantly improve the spatial coverage of surface $pCO_2$ and could improve seasonal bias, but would not have a significant impact on our ability to resolve the full carbonate system or to consider deeper water. Glider datasets would similarly improve our spatial coverage while providing additional information about water column structure. These could represent a valuable expansion to this synthesis, provided calibration records are also available, and will likely be included in updates to this synthesis product (Bushinsky et al., 2019). Shore-based monitoring systems recently deployed by the West Coast OOIs would be valuable expansions to this synthesis and will also likely be included in an updated product.

## 4 Conclusions

The CCS is one of the most intensively monitored marine systems in the world, but our ability to accurately resolve the true complexity of coastal climate stress remains limited by data fragmentation, availability, and quality. As interest has shifted from documentation of the global patterns of acidification and hypoxia to more complex coastal environments, the CCS has seen an explosion in monitoring efforts within 50 km of shore in the last 15 years. This expansion has included an increase in both surface and subsurface monitoring efforts, though within 2 km of shore, monitoring efforts below 5 m depth are still much less common than surface observations. While this situation is improving, the continued relative paucity of subsurface nearshore measurements is of particular concern given that mildly hypoxic (DO < 107 µmol kg$^{-1}$) and corrosive conditions have been documented at depths as shallow as 10 m (Kekuewa et al., 2022).

Surprisingly, the U.S. West Coast had especially continuous spatial and temporal coverage of OAH-relevant parameters between 2012 and the beginning of 2015, before a reduction in coverage that lasted through 2020 (Fig. 5). By coincidence,

the reduction in DO and carbonate system monitoring in 2015 coincided with the second half of the marine heatwave known as "the Blob", which stretched from 2014 through 2016 and was associated with higher surface DO and pH (Bond et al., 2015; Siedlecki et al., 2016; Gentemann et al., 2017). Assessing the interactions of an unprecedented marine heatwave with DO and carbonate system conditions lies at the heart of multistressor risk management; however, our ability to resolve both Blob impacts and its recovery was very limited in Northern California and Oregon by the concurrent contraction in oceanographic monitoring. Although the CCS is well monitored compared to many other parts of the world's oceans, our synthesis here highlights that the patchiness of monitoring projects, often driven by inconsistent funding, has an outsized impact on our ability to utilize those data to operationally monitor for climate change.

While increasing interest in coastal OAH monitoring and the availability of autonomous sensors has markedly enhanced CCS data availability, the frequency and footprints of synoptic oceanographic cruises has decreased in the region. Oceanographic cruises provide highly accurate and spatially broad water column measurements that can bridge the gap between the coastal and open-ocean domains and provide regional contexts for local observations. They also provide some of our only observations near remote portions of the coast. However, nearly all routine oceanographic cruises in the CCS have cut back their footprint, sampling frequency, and depth resolution. The Southern California-based CalCOFI cruises extended throughout the CCS during the 1960s, contracted to Southern and Central California by the 1980s, and now only covers the Southern California Bight while also sampling at significantly fewer depths (Bograd et al., 2003). The loss of CalCOFI cruises in Central California has been offset in part by triannual Applied California Current Ecosystem Studies (ACCESS) cruises near San Francisco Bay, though these cruises are limited to the continental shelf between 37.3° N and 38.4° N. The NOAA West Coast Ocean Acidification (WCOA) cruises took place along the entire CCS five times from 2007 to 2016, but did not occur again until 2021 (Feely et al., 2016a; Feely et al., 2022). The shift towards high-resolution, nearshore monitoring is a significant improvement over a wholesale reduction in oceanographic monitoring, but the concurrent erosion of consistent oceanographic cruises means the ability to resolve large-scale regional patterns is being traded for highly specific understanding of a few select locations.

This synthesis dataset provides one of the largest compilations to date of West Coast nearshore acidification- and deoxygenation-related data. This dataset highlights monitoring gaps, but equally provides opportunities for insight into coastal conditions. With the updated spatiotemporal resolution our effort affords, this dataset offers a wealth of opportunities to investigate questions about coastal oceanography and evaluate localized patterns of marine climate stress. We expect the MOCHA synthesis to also be of use for new projects combining temperature and DO records into species metabolic indices (e.g., Howard et al., 2020b), for investigating the frequency and interaction of individual and overlapping ocean acidification and hypoxic events (e.g., Burger et al., 2022), and for developing updated carbonate system algorithms more suited to coastal environments (e.g., Alin et al., 2012; Davis et al., 2018). By archiving this dataset at the National Centers for Environmental Information (https://doi.org/10.25921/2vve-fh39; Kennedy et al., 2023) in an easily manipulated, consistent

format that includes relevant metadata and quality assurance, we provide an important tool for scientists across ecological, oceanographic, and social disciplines as well as coastal decision-makers to address the environmental, economic, and cultural needs of coastal communities.

## 5 Data Availability

The full Multistressor Observations of Coastal Hypoxia and Acidification dataset, parameter metadata, and dataset metadata tables are publicly available for download at NCEI as Accession 0277984 with the DOI 10.25921/2vve-fh39 (Kennedy et al., 2023). The downloadable content includes the full MOCHA dataset available as a text file, the daily summarized dataset discussed extensively above available as a text file (aggregated_daily_dataset.csv), standard NCEI accession parameter metadata which provides an overview for each variable included in the text files ("SubmissionForm_carbon_v1_428.xlsx"), and a bespoke dataset metadata table describing each included dataset with citations and links to reference papers (MOCHA_dataset_metadata_table_v2.csv). This data package is discoverable via the NOAA Ocean Acidification Portal, NCEI Geoportal (https://www.ncei.noaa.gov/metadata/geoportal/#searchPanel), and other online discovery tools. The dataset metadata table is also available in the Supplemental Information for this paper.

## 6 Fair Use Data Statement

We request that all users of the MOCHA compilation also fully credit the constituent datasets supporting their work. This helps ensure that the ocean monitoring systems that this, and other, compilations depend on receive trackable citations and continued funding. We also recommend contacting the original principal investigators to discuss collaboration opportunities and to enthusiastically look for opportunities to further include or credit these data providers. Full citation information, dataset DOIs, and reference papers (where available) for each individual dataset in the MOCHA compilation can be found in the References as well as in the MOCHA_dataset_metadata_table.csv available at NCEI (https://www.ncei.noaa.gov/data/oceans/ncei/ocads/data/0277984/).

## 7 Code Availability

Code for performing carbonate system calculations with the formatted dataset, creating a summarized dataset aggregated by day, and making all included figures is available on GitHub at https://github.com/egkennedy/DSP_public_code.

## Competing Interests

The authors declare that they have no conflicts of interest.

## Author Contributions

After the first four contributing authors, additional authors are listed alphabetically in two groups: those who contributed significantly to data acquisition, interpretation, and overall project direction and those who contributed to data curation. All authors read, edited and approved of the manuscript. EGK wrote original draft and led data curation and quality control methodology. MZ and SLH provided substantial manuscript reviews, data curation, and methodology insights. TMH led project conceptualization, funding acquisition, and supervision, and provided substantial manuscript review. TMH, KJK, JJ, CF, and ME provided previously unpublished data for inclusion. KJK, AKS, BG, ES, and MW contributed to funding acquisition and project conceptualization. HMP, MW, AMR, GVG, CNR, GC, MD, MIW, EH, and SW provided data curation and sourced new datasets for inclusion.

## Acknowledgements

This project received funding from the following sources: the California Ocean Protection Council (Hill, Sanford, Gaylord, Kroeker, Spalding), Lenfest Ocean Program (to Hill, Sanford, Gaylord, Kroeker, Spalding), NOAA Ocean Acidification Program (to Hill and Spalding), Packard Foundation (Hill, Sanford, Gaylord, Kroeker), and National Park Service (to Hill) and National Science Foundation (to Kennedy, Grant No. 1734999). We value the partnership of the Applied California Current Ecosystem Studies Partnership (ACCESS, www.accessoceans.org), an ongoing collaboration between Point Blue Conservation Science and Greater Farallones and the Cordell Bank National Marine Sanctuaries to support healthy oceans in north-central California. Brady O'Donnell, Priya Shukla, Lena Capece, Seth Miller, Sarah Merolla, Daphne Bradley and Danielle Lipski assisted with data collection and management. We thank Rachel Carlson and Alyssa Griffin for sharing their advice, expertise, and feedback at key points in the project. We also thank the UC Davis Data Lab for their assistance with data management and coding.

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
