# Peer review of "A high-resolution synthesis dataset for multistressor analyses along the U.S. West Coast"

_Earth System Science Data, 2023_

## Author Comment (AC1)

**Author note:**
We thank all referees for their insightful and constructive comments on our manuscript "A high-resolution synthesis dataset for multistressor analyses along the U.S. West Coast." We appreciate the opportunity to incorporate and respond to these thoughtful comments and improve our manuscript. Below, we discuss the comments from Reviewer 1. We have included all original comments, with our response to each point raised bulleted below.

**Rev 1**
This manuscript is a well-written description of a data compilation/synthesis effort for the California Current System (CCS). The main value added over existing data products is: 1) the inclusion of nearshore data sets that are missing in many of the larger-scale data compilation/QC products, and 2) the inclusion of data sets that explicitly address CCS temperature and O2 data along with the carbonate chemistry data that are the focus of the larger-scale data products I'll discuss further below. I think these two things make this data compilation a valuable contribution that will advance the state of coastal multi-stressor work in the CCS after the paper's weaker points are adequately addressed. However, I am a bit challenged by several aspects of the paper and data set in their current form.

**Major concerns:**

For starters, modelers and other scientists doing ocean acidification-related coastal analyses would likely still be best advised to use the existing Surface Ocean CO2 Atlas (SOCAT.info, see Dorothee Bakker et al. 2016 ref in ESSD) and Coastal Ocean Data Analysis Product for North America (CODAP-NA, see Li-Qing Jiang et al. 2021 ref in ESSD), which have compiled all global surface ocean CO2 data and coastal carbonate system data for North America, respectively. In my view, both of these projects provided a more rigorous secondary QC of the data, or at least a more detailed description of the secondary QC processes, while also providing the other benefits of the MOCHA data effort (consistent formatting and data treatment, etc.).

- We appreciate this reminder to highlight SOCAT and CODAP-NA, as both are admirable data compilations. SOCAT and CODAP-NA are exceptional products for oceanic geochemists and modelers, but neither product is ideal for nearshore-focused carbonate chemistry and hypoxia work. SOCAT provides surface $CO_2$ data and includes many nearshore records, but lacks deeper records and other carbonate-system parameters (e.g., TA, DIC, and pH). For nearshore

work focused on synoptic cruise observations, CODAP-NA is an excellent resource with high-quality data, but it excludes many coastal records such as moorings and shore observations. We feel there is a need for a synthesis product that combines the consistent formatting and organization of SOCAT and CODAP-NA with the data inclusivity of regional databases such as that maintained by CenCOOS, allowing coastal researchers to examine cruise and sensor observations in tandem. We have added language to the fourth paragraph of our introduction (lines 92-111) highlighting the strengths of SOCAT and CODAP-NA while clarifying the value we see in bringing sensor and cruise datasets together.

This makes me wonder who the envisioned target end user group is for the MOCHA data products beyond scientists. To be clear, I can certainly see broad utility for the data, as amassed here, for the added nearshore data and T and O2 data.

- We envision this dataset being of particular interest to coastal scientists working on or supporting policy- and management-relevant research questions. We have added the following to the fifth paragraph of the introduction to clarify this (lines 128-134): "We anticipate that this synthesis product will be broadly useful to OAH-focused investigative teams and particularly impactful for coastal scientists investigating policy- and management-relevant projects, such as investigating spatiotemporal variation in marine climate risk from OAH events and warming, evaluating the efficacy and completeness of CCS monitoring efforts, linking oceanographic conditions to coastal social or socio-economic considerations across large geographic ranges (e.g., Ward et al., 2022), evaluating spatial management zones such as aquaculture sites and marine protected areas (e.g., Hamilton et al., 2023), and pursuing other questions of interest to coastal communities."

But if the end users are not scientists and capable with programming, I do think it might be useful to provide some additional data products that would be easy to create and archive at NCEI and much more accessible to less technically savvy end users. I say this because I was unable to open the nearly 3 GB data file in Excel on my computer. I was able to open it in R, but it was still very slow and cumbersome to use there.

- The size of the dataset is a challenge and certainly limits its utility to non-scientists. We agree with your concerns and, to that end, we have been collaborating with the Central and Northern California Ocean Observing System (CeNCOOS) to

incorporate the data into their web based portal. Additionally, we are posting the aggregated daily dataset to NCEI in accordance with your suggestion below.

SOCAT and CODAP-NA provide a useful model for one way I'd imagine you might subset this data product: 1) one product including observations that reflect the surface conditions (where I would likely use 10-25 m as the depth cutoff, rather than 50 m as they used for their TA-S analysis), and 2) another smaller subset of observations that were depth-resolved and included a broader range of parameters (nutrients, chl, etc.). This would eliminate the need for millions of empty/"NA" entries, that I presume slow down operations with the data set in R (and make it inoperable in Excel [in my experience]). Further, the authors provided this massive data compilation, and the code they used to create the greatly reduced data summary, around which much of the discussion was written. I would suggest also directly providing this summary data product (along with the surface and full water column data subsets) via the NCEI webpage where the original data set is logged.

- We appreciate the suggestions for some more user-friendly data compilations. We have uploaded our daily summary dataset to NCEI as suggested (aggregated_daily_dataset.csv). At 1.2 million rows, this aggregated dataset is still slightly too large for easy handling in Excel, but it is much easier to work with than the full dataset. While we appreciate the suggestion for additional gridded surface and depth-resolved data products à la SOCAT and CODAP-NA, summarizing the MOCHA dataset over a spatiotemporal grid would require significant interpretive decisions that are better handled by individual science teams. For example, if a given spatial area contains several autonomous sensors from different projects with different protocols along with data from one or more cruise stations, how should the data be aggregated and combined? The correct protocol in this case depends on the project and question asked and is beyond the scope of this paper.

I do also think that the authors need to discuss this data product in relation to the SOCAT and CODAP-NA data products and how they are related and pros/cons of each. This comment got long, so to recap, what I'd like to see addressed here are: 1) providing alternate data products to facilitate accessibility for various end users, including possibly splitting out surface and depth-resolved data product subsets and providing the data summary directly; 2) put this data product into the context of existing high-quality data products like SOCAT and CODAP-NA.

- Please see our detailed response to this suggestion in the section above.

Second, as a person whose livelihood comes from producing data sets such as those included here, it was disappointing to not see reference to data providers and original data set DOIs (for those that have them) in Table 1 in the main paper. Yes, this information is in the very similar metadata file at NCEI, but in my experience, no one reads the metadata, so the major amount of work data providers do is not going to be appreciated/cited/acknowledged. It is important for the on-going funding and ability to do observations that data producers are able to find and report papers that rely on their data for subsequent publications. On a related note, I completely agree with the authors that there is a significant need for not just continued observational coverage, but expanded observational coverage, particularly for the carbonate system, in a future world with accelerating rates of change, marine carbon dioxide removal, etc., etc. To that end, it is critical that data creators get fair acknowledgement of their products.

- We completely agree, and apologize that our misunderstanding of what could be included in the References section meant that we did not include dataset citations or DOIs in Table 1. We have since added those following an excellent example from Sutton et al., 2019 (https://doi.org/10.5194/essd-11-421-2019). Table 1 and the References section now both include full citations for each dataset. Now that the datasets listed can be identified by their citations, we have also shortened their titles and improved the overall readability of Table 1.

Consequently, it would be ideal to see all of the data sets appear as citations in the main article of this paper. I also encourage the authors to consider including a "fair data use statement" in their data availability section regarding the data product. They can see the SOCAT statement here: https://socat.info/wp-content/uploads/2023/06/2023_SOCATv2023_Data_Use_Statement.pdf. And the GLODAP statement here (GLODAP is the open ocean data product that CODAP-NA was modeled after): https://glodap.info/index.php/fair-data-use-statement/

- A fair data use statement is an excellent suggestion. We have added one just after our Data Availability section requesting that users fully credit constituent datasets and reach out to original PIs, as appropriate (lines 640 to 646) .

To recap, 1) it would be nice to see better inclusion of main data provider information and DOIs in Table 1, along with citations for all data sets in the main manuscript if

possible; 2) I encourage the authors to consider adding a "fair data use statement" that would encourage end users to cite both the MOCHA paper/data set AND authors of any major subset of the data used for follow on publications and information products, as appropriate, to help support the long-term stability of observational programs.

The TA-S analysis was not fully described or discussed. It was unclear why they would have used the upper 50 m rather than a smaller part of the upper water column, as other authors have done. Maybe they determined this experimentally, but how they arrived at this decision should at least be described. However, if one is expecting to discern the influence of freshwater, this should likely be a shallower depth range. Further, there were some really strange results—e.g., offshore of SF Bay mouth—that were not discussed adequately. Also, these results were not placed in the context of other publications by Andrea Fassbender (and references therein) or Kitack Lee.

- We appreciate this reviewer's (and Reviewer 3's) questions about these data, as we had also been puzzled by them. In light of these questions, we went back to the raw, original titrator files and realized there was a years-old issue with quality control on some autotitrator runs associated with CRMs that had not run well. In total, these autotitrator quality control issues impacted a subset of three of the author team-provided datasets in the MOCHA compilation. All of the impacted total alkalinity measurements and their associated samples have now been entirely removed from the uploaded dataset and the figures and led us to substantively rewrite this section (new text in lines 485-515). The remaining variability in coastal total alkalinity comes from samples that have been thoroughly examined. This variability is in line with previous investigations, plausible given riverine inputs and potential organic alkalinity contributions, limited to within 2 km of shore, and no longer shows San Francisco Bay as an anomalous region. We have removed the previous Figure 7 and Table 3 with a revised Figure 7 showing regional surface (< 25 m depth, as suggested) TA-S relationships for coastal (< 2 km from shore) and offshore (2-100 km from shore) zones. We have also placed these observations in context with Fassbender et al. (2017) and others.

Along these lines, and because of the importance of salinity data to the carbonate system (as reflected by the TA-S work discussed just above), I will note that mention of salinity felt a bit inconsistent throughout (e.g. one place it was noted that DO and pH data were not included if they did not have accompanying temperature data, but it left me wondering—what about salinity data (lines 135-136)?

- We evaluated pH data from ISFET and spectrophotometric sensors that were not accompanied by salinity measurements on a case-by-case basis. Where high-quality pH data passed all other QC checks (e.g., diver-accessed bottom sensors used in Donham et al., 2023 and coastal monitoring by the OMEGAS program, e.g., Chan et al, 2017), we retained the pH data. This detail has been added to our Methods section (lines 174-176).

Also on line 298 where T and DO are mentioned as having the widest coverage—presumably also S? I ask because low S events associated with flooding may also be associated with coastal multistressor events (e.g., potential importance in kicking off HAB events).

- Thank you for the suggestion; the links between low salinity and HAB events represents an application of our dataset that we hadn't fully appreciated. We have added text to clarify how the amount and spatiotemporal coverage of salinity data compares to the figures shown, as salinity observation density hews closely to dissolved oxygen observations in both spatial and temporal coverage (lines 386-387).

Finally, the "detailed metadata" file referred to in the text at NCEI I think is the one actually called "SubmissionForm_carbon_v1_428.xlsx" and is NOT the one called "MOCHA_dataset_metadata_table.csv". This is confusing and should be clarified by either adding the name of the actual file intended to be referenced here the main text (probably in parentheses) or by asking NCEI to rename the file at NCEI to "detailed metadata..." (or whatever the final name used in the manuscript is).

- This is an excellent point. We actually intended the "detailed metadata" to point toward the MOCHA_dataset_metadata_table.csv. All references to "metadata" in the text have now been clarified to refer explicitly to either the "dataset metadata" (MOCHA_dataset_metadata_table_v2.csv) or the "parameter metadata" (SubmissionForm_carbon_v1_428.xlsx). The following sentence has also been

added to our Data Availability section: "The downloadable content includes the full MOCHA dataset available as a text file, the daily summarized dataset discussed extensively above available as a text file (aggregated_daily_dataset.csv), standard NCEI accession parameter metadata which provides an overview for each variable included in the text files ("SubmissionForm_carbon_v1_428.xlsx"), and a bespoke dataset metadata table describing each included dataset with citations and links to reference papers (MOCHA_dataset_metadata_table_v2.csv)."

**Less major concerns:**

On a positive note: I do like the simple data QC flagging routine they used. If a major portion of end users are non-technical, this will greatly facilitate the uptake and correct use of this data product. That said, another benefit of directly providing the data summary product, beyond its vastly smaller file size is that it only includes the "reliable" data. So non-technical users should definitely be steered toward that sub-product.

- Thank you, we appreciate this point. While the QARTOD flags are more detailed, the variety of data sources and previously applied QA/QC practices we worked with in this synthesis pointed toward applying a simpler system.

It would be nice to use the recommended/best practices column headers recommended by Jiang et al. 2022.

- We found those headers and dataset structure to be less appropriate given the mixed nature of our data sources (cruises, shore samples, autonomous sensors, etc.), whereas the Jiang et al., (2022) headers are very well tailored toward either discrete cruise data or moorings. We have uploaded code to our Github repository to convert our column headers and dataset into a format more compatible with Jiang et al.'s recommendations, for those who would prefer ("reformat_toward_NCEI_standard.R").

--Along similar lines, Jiang et al. 2022 recommend using different carbonate system coefficients and would be worth a look for future use. I do not believe there would be a noticeable difference in your results, so am not necessarily suggesting you re-do anything here, because you don't submit or show the calculated parameters.

- Since the inclusion of calculated carbonate system parameters did not meaningfully change our figures or paper conclusions, we have entirely removed unmeasured pH values from our paper, so this suggestion is moot but well-taken. As there are no longer any calculated parameters in our figures, we also have removed section 2.6 (Additional Carbonate System Calculations).

Jiang et al 2022 also point out that units of µmol/kg refer to "substance content" rather than "concentrations," which are in µmol/L units. This should be corrected in the "Submission form_carbon_v1_428.xlsx" at NCEI and in the text as well.

- This has been fixed throughout and at NCEI. Thank you.

In Table 1, is #68 a gridded data set? I got that impression, and if so, I'd argue it's not appropriate to include here. The language should be clarified around this.

- Thank you for noticing this apparent error. Dataset 68 is actually composed of the CTD casts used to create the "gridded dataset" referenced in the title. We have fixed the title and description of this dataset to make it clear that we are using the CTD cast data that Risien et al. (2022) then used to create their gridded data product.

--It would be useful to state more decisively in the early text that the data were limited to within US border. It's alluded to a few times, but because I happen to know that some of the data sets span the Canadian and/or Mexican border, as does the CCS, I didn't initially catch it. Easy enough to justify.

- This has been added to both our Introduction and early in the Methods section (lines 118 and 158).

I don't believe they mentioned which pH scale they used in the text, although it is in the "Submission form" file. Please add to the text, and for any original files that used a different pH scale, whether/how they converted to the same scale.

- We have clarified that we are using the total pH scale in our methods section. Surprisingly, no pH scale conversions were required during this compilation. Along the same lines of this comment, though, we have added details about converting pH measurements to in-situ conditions when necessary to our Methods section (lines 209-212).

In Table 1, ship names should be italicized and 2s in CO2 or O2 should be subscripted.

- This has been fixed.

--Finally, as noted previously, I completely agree with the authors about the importance of the coastal multi-stressor observations, and particularly carbonate system observations, needing to be sustained or expanded rather than contracted, but there was an incorrect statement in the conclusions section regarding the NOAA West Coast Ocean Acidification (WCOA) cruises. Unfortunately it's also mislabeled on the NCEI WCOA web page here: https://www.ncei.noaa.gov/access/ocean-carbon-acidification-data-system/oceans/Coastal/WCOA.html

Specifically the 2017 cruise was not a WCOA cruise. Rather it was a collaborative effort led by NOAA HABs scientists and an added cruise-of-opportunity for OA sampling. I think they also sampled OR on that cruise, but I haven't looked at the data for a long time so the authors should double check this (the title said PNW, so I assume Oregon was included). However, NOAA did have another full US West Coast OA cruise in 2021. It was delayed from 2020 due to COVID. Thus, please edit that sentence to not state that WCOA cruises have contracted.

- We appreciate the correction and tip toward the 2021 WCOA cruise. We have updated the title of dataset 41 to "NOAA Northwest Pacific harmful algal bloom program cruise SH1709" in Table 1. We have also updated our discussion of synoptic oceanographic cruises to note that there has been a WCOA cruise since 2021 (line 609).

**Minor concerns:**

- Unless otherwise noted, all following suggestions have been fully incorporated.

--The DOI in the abstract doesn't go to the data set.

- The DOI seems to be correct, but we have re-checked this.

--Figure 1—why are the a and b panels smaller than c? It lends some confusion when all could be the same size  and fit nicely across the page.

- This figure has been updated and all maps are now on the same scale.

--I don't think "carbonate system" needs to be hyphenated. I am familiar with how this works with adjectives vs nouns, but it is not used consistently throughout the manuscript in any case. Also, there was at least one place where one might hyphenate dissolved oxygen where carbonate system was hyphenated (e.g., line 119).

- We have eliminated all hyphens in "carbonate system" throughout the manuscript. Thank you.

--L. 123—specifies data collected *before* 2020 but there are at least two places in the dataset metadata table that say either 2020 or 2021.

- These typos have been corrected at NCEI.

--dataset ID 2 in Table 1 and in the Excel metadata table—There's a space before the text that makes a gap appear in the excel file.

--dataset 5 in Excel file—Greeley is misspelled

--Table 1 dataset 25—should be to 2020 not "present"

--dataset 41—didn't this also include Oregon? (It says Pacific NW)

--dataset 52—La Push is two words

--dataset 68—again, the words "gridded" and "monthly climatologies" make me think this data set may not be right for inclusion in MOCHA

- Please see our more detailed comment about this above. This language and the title of this dataset have been clarified to reflect that we used CTD cast data.

--Lines 224-226—It might be useful to differentiate between the # of samples dropped as questionable data vs. those dropped due to daily averaging, because this sentence gives the impression that there were more 3s than there were.

- This point has been clarified. We now explicitly note that the reduction in data quantity is not a result of unreliable data, but simply a result of collapsing high resolution sensor datasets into daily summaries (lines 289-290).

--line 232—I'm not sure that "high-quality" was defined anywhere. Uncertainties definitely were not adequately spelled out across the data sets, and I'm almost certain the

uncertainties would have varied across the 71 distinct data sets used. This information doesn't seem to be in either the submission form or the metadata file on NCEI.

- We have replaced "high-quality" as a term with "plausible and reliable" data, versus data that is "unreliable". We have further clarified in our Quality Control section (2.4) that our "plausible and reliable" data may warrant additional QC depending on the investigator's needs. The uncertainty of datasets does vary significantly throughout this compilation and, in many cases, was not available to us with the published data. Our intent was to clear the compiled data of all unreliable observations, but we can not assert that all the "plausible and reliable" data meets an objective accuracy or precision standard.

--I encountered some confusion between "handheld" sensor measurements vs. those collected "by hand" (hand collected—line 155)—maybe making the latter not use "hand" would prevent others' confusion when thinking back to what earlier categories of observations and instruments were.

- Excellent point. We have changed the sampling scheme "intertidal/subtidal hand collected" to be "intertidal/subtidal discrete collection".

--L. 162—TA is not "extrapolated" from S measurements—please reword

--Throughout—the word "data" always gets a "plural" verb tense

--Table 2: missing value in reliability column for calculated pH

- There is now no calculated pH data in the paper so this row has been removed.

--Lines 296-297—Please indicate on the figure where Pt Arena, CA, and central OR are for readers' convenience. It could just be asterisks along the axes or similar.

--Line 312—Should say July through September (it's correct in the figure caption, but the caption doesn't include May, which it should).

--I liked the discussion of the co-occurrence of stressful DO and pH conditions—I have been looking at similar occurrence statistics myself. And I agree with the conclusion about this pointing to a need for expanded CO2 system observations. It may be useful in this discussion to give DO results in alternate units also (mg/L and mL/L) for our colleagues and end users who use different units.

--Figures 5 and 6 (and elsewhere)—again, should be DO content rather than concentration

--Line 382 and Table 3 caption—the p values do not agree.

--Table 3—again, the offshore relationship with the r squared of 0 seems to require further explanation than given. Specifically, while I would buy that the effect of urban runoff could be strong outside SF Bay, none of the #s in the offshore box make any sense—they are all SO different from all other boxes, including the nearshore SF Bay one, that it makes me wonder if there was an error in the analysis or a typo.

- Please see our detailed response to this observation in the section above. Table 3 has now been entirely replaced by the new Figure 7 and the conclusions therein are much more compatible with anticipated offshore vs. onshore TA-S relationships.

--Lines 418-421—Really seems like the authors are not aware of the wealth of surface CO2 data in SOCAT. This is one of the places where SOCAT might be drawn into the discussion.

---

## Author Comment (AC2)

**Author Note:**
We thank all referees for their insightful and constructive comments on our manuscript "A high-resolution synthesis dataset for multistressor analyses along the U.S. West Coast." We appreciate the opportunity to incorporate and respond to these thoughtful comments and improve our manuscript. Below, we discuss the comments from Reviewer 2. We have included all original comments, with our response to each point raised bulleted below.

**Rev 2**
This paper documents the development of a large dataset of observations of dissolved oxygen, pH (and other carbonate chemistry parameters), and temperature in addition to a few other low priority ad hoc variables (e.g., nutrients). The dataset will be very useful to the broader scientific community and the paper is generally well-written. I inspected the data posted the public repository and it is in excellent shape. I think the paper is ready to be accepted after some minor comments listed below.

**Major comments**

The only thing approaching a major comment is that I got confused about the number of observations in the data. At one point in the methods, it sounds like the aggregation of the data into daily averages reduced the dataset from 12.7 million rows to 1.2 million rows but then in the results it sounds like there are 12.7 million rows and the aggregation wasn't done. Please be very careful about this and report accurately how many observations are in the final (data available to user) dataset and propagate throughout.

- Thank you for bringing our attention to our confusing wording. We have now explicitly defined an "observation" in our Methods section to be a row in our dataset, in which one or more individual parameter measurements will be associated with a dataset, date, time, and location (lines 191-196). We have further replaced "measurement" with the more specific "individual parameter measurement" to make it clear when we're discussing the amount of data associated with a single parameter (e.g., 13.7 individual temperature measurements, but only 3.3 million individual dissolved oxygen measurements). Additionally, we have clarified the Results section discussions of the aggregated daily dataset by breaking the previous section "3.1 Overall Data Totals" section into two smaller sections, "3.1 Overall Measurement Quantity" and "3.2 Aggregated Daily Data Totals."

The metadata table (MOCHA_dataset_metadata_table.csv) is easy to understand and seems largely complete though dataset 2 does not have a name. The other fields missing data make sense.

- We have fixed the error with dataset 2. We are pleased to hear that the dataset metadata table is easy to interpret!

I examined a subset of the data ("47_to_49_pre_2015.csv") and it is in great shape. The column names are all super intuitive and the values in the columns are all correctly formatted. All of the data that you would expect to be complete is complete. Wonderful.

Minor note for future submissions: please use continuous line numbering (not 5 line intervals). Do everything you can to make the reviewers job easy – this will keep them happy!

- We appreciate this feedback and have passed it on to ESSD for potential incorporation into their manuscript templates.

**Minor comments**

- Unless otherwise indicated or discussed, all of the following minor comments and suggestions have been fully incorporated. We appreciate the reviewer's close attention to detail.

Abstract

24 – Stressful or favorable, plus what's stressful for one organism might not be stressful for another

31 – could you work the focus on hypoxia and ocean acidification risk a little earlier in abstract?

32 – stats on the time span of observations should get mentioned

Introduction

43 – could shorten "effluent from coastal settlements and agriculture" to "coastal runoff"

43 – it's not clear to me the mechanism for "diverse and highly productive ecological communities" to drive local deviations from global patterns

- Here, we were referencing how local biomes like seagrass meadows and kelp forests can significantly alter the local chemical environment (e.g., Ricart et al., 2021). We have changed this phrase to "high local productivity".

52 – "e.g." is missing a comma after it (like in line 40); ensure comma is added throughout

61 – Free et al. (2023) (https://doi.org/10.1111/faf.12753) provides an update to Cavole et al. 2016 paper and is explicitly about this region

71 – Can you make it clear that conditions have gotten shallower without using the word "shoaled"? It might not be familiar to everyone.

89 – What regions do they apply to?

- The specific references to CenCOOS and SCCOOS have been removed..

97 - "…for the CCS and is newly archived and available at…"

102 – Hoping to see that the unincorporated sources of info get mentioned later

107 – no need to capitalize MPAs

108 – The stats on number of observations, sources, and time span should get mentioned in last paragraph of info

Methods

119 – how was the literature search conducted?

- We did not do a formal literature search to find datasets and meant here to refer to sourcing some datasets from published literature. More accurately, we accessed public data portals and federal government datasets, contacted colleagues to request their assistance in locating datasets, presented the project

at conferences for three years requesting community participation in the project, and completed a scan of published literature that likely included published datasets that could be incorporated into the project. While this is not considered exhaustive, one benefit of the NCEI data platform is that we can continue to update the available data as we become aware of and process new sources. We have updated our description of sourcing datasets to this more detailed description (lines 150-153).

138 – suggest adding (1), (2), (3), and (4) here to orient reader

153 - suggest adding (1), (2), (3), and (4) here to orient reader

158 – this should at least be a supplemental table in this paper; its annoying to have to go look elsewhere for info on the dataset documented in this paper

- The "dataset metadata table" has now been included as a supplement in addition to being available at NCEI.

191 – What does "as normal" mean here?

- This phrase has been altered to "quality controlling them further following the practices described for all other incorporated datasets."

199- suggest adding (1), (2), (3) here to orient reader

205 – Can you give examples in the supplement? Reads as vague now

- We have added an example of our formatting and flagging practices as supplementary information and have made the code and data associated with it available on our project Github repository (github.com/egkennedy/DSP_public_code).

210 – Examples drawing from this would be useful

- We believe this request has been addressed through our supplemental flagging example.

224 – "i.e." should be followed by comma – correct throughout

Results and Discussion

244 – Again, time range would be helpful.

244 – Isn't 12.7 million incorrect? Didn't you reduce down to 1.2 million by aggregating to daily level as stated in Line 226. I'm skeptical of all the sample sizes reported here b/c of this.

- Following our response to the reviewer's major comment above, we have revised this paragraph to make it clearer that we are discussing the full, disaggregated data set here, rather than the aggregated dataset we used for oceanographic interpretations. The aggregated dataset is now discussed in its own subsection just below.

273 – "malfunction, 2)"

273 – I think either means between two options

Conclusion

474 – No need to capitalize MPAs

Tables and Figures

Figure 1. The figure would be more useful if it showed the density of points along a raster grid (potentially hexagonal) so that the reader understands data density spatially. The panels should all be the same size, 1 row, 3 columns would be an improvement. The density could be the number of points within a cell or the number of unique year-months in a cell. I leave it to the authors.

- We appreciate this suggestion. There is inherent tension between showing every available data location and the data density over space or time. We believe both are valuable, but have kept this figure showing all of the individual data locations since we show the spatiotemporal data density more clearly in Figure 3. We have taken the suggestion to make all maps in this figure the same size.

Figure 2. Y-axis is a proportion, not a percentage. Align the word choice with what is shown. Spell out acronyms in caption.

Figure 3. It would be nice if the panels were labeled with the parameter so the reader doesn't even have to read the caption. The width of the latitude should be stated.

Eyeballing the figure. Data looks to be most common between 2015-2020 and not 2010-2015 at the authors state, Bar plots of annual totals would be a good way to examine the temporal bias alone.

- We have added parameter titles to the plots and updated the caption to clarify that the spatial coverage of observations is most complete between 2010-2015, whereas the total number of observations is highest between 2015-2020.

Figure 6. The caption is confusing about what the points are. Are these all observations with 50 km of shore in the top 50 m? State what it shows. Currently, it's written like a results section. Define the lines but exclude all of this results interpretation.

Figure 7 caption also includes lots of results interpretation.

- Both the captions for Figures 6 and 7 have been rewritten to exclude results interpretation. Figure 7 has also been substantially revised and now replaces Table 3.

Figure 4. Y-axis should read "Percent of observations."

Table 1. Define acronyms in parameters column in caption. Consider making this a supplemental figure given its size.

- While Table 1 is long, it was important to us to give credit to the constituent datasets of this synthesis compilation so we hesitate to bury it in a Supplement. We have added a DOI/Citation column to the table and full references to all datasets to our References section. This has allowed us to simplify the titles in Table 1 and improve the readability of the table, though it is admittedly still quite long. We have defined the parameter acronyms as suggested in the caption.

Table 2. Add comma to 3rd column. Eliminate 2nd decimal spot in fourth column. Spell our Parameter acronyms in caption.

Table 3. This would be more compelling as a multi-panel figure of scatter plots with regression fits. Caption is mostly results interpretation.

- Thank you, we have replaced Table 3 with this suggested figure and removed the previous Figure 7.

---

## Author Comment (AC3)

**Author Note:**
We thank all referees for their insightful and constructive comments on our manuscript "A high-resolution synthesis dataset for multistressor analyses along the U.S. West Coast." We appreciate the opportunity to incorporate and respond to these thoughtful comments and improve our manuscript. Below, we discuss the comments from Reviewer 3. We have included all original comments, with our response to each point raised bulleted below.

**Rev 3**
Kennedy et al. collate, quality control, and synthesize temperature, salinity, and biogeochemical data in the nearshore region of the U.S. portion of the California Current Ecosystem. This data product does show promise for addressing temporal and spatial variability and multistressor dynamics within this region, however, the associated manuscript does not provide enough information for a potential data user to fully understand the appropriate applications for the data product or how the data are manipulated. It also does not provide fair credit for the contributions of the original data providers and funders.

**Major comments:**

The authors claim the science applications of this data product are broad, including characterizing seasonal variability and spatial variability along the U.S. West Coast. However, the results illustrating variability only focus on the portion of the data sets within 50 km of the coast and < 25 m depth. Either the results need to be expanded to include analysis of the entire data product, or the data product should be restricted to the shallow, nearshore environment and the title and introduction should reflect that the product is focused on the nearshore.

- It is our intention to provide a data product that lends itself to a wide range of spatial and temporal scientific questions, rather than limiting our data compilation to the use of the specific case studies we discuss in this paper. We imagine future research using the MOCHA synthesis that defines its areas of interest via socioeconomic boundaries such as the U.S. Exclusive Economic Zone, bathymetric boundaries such as the region shoreward of the continental shelf break, ecological boundaries such as viable kelp habitat, or other bespoke regions that extend from the shoreline. We have added language to paragraph 5 of our introduction to frame our shallow, nearshore examples as examples, rather than exhaustive analyses (lines 120-126)

Given the data set itself is not interoperable or compatible with other products that include offshore biogeochemical data (e.g. gridded NetCDF files of the Surface Ocean CO2 Atlas or Biogeochemical Argo), it would be difficult for a user to combine and utilize them for assessment of biogeochemistry spanning offshore to nearshore.

- We appreciate the reviewer noting the lack of interoperability between many coastal and oceanic datasets. Even without considering interpolated gridded products, the difficulties of working with data from cruises, moorings, and shore samples was a large part of the motivation for developing this synthesis. Existing coastal syntheses such as SOCAT or CODAP-NA are highly specific – only surface $CO_2$ measurements in the case of SOCAT and only oceanographic cruise discrete samples in the case of CODAP-NA. These are excellent products, but very difficult to augment with the wealth of high resolution sensor data also available in the region. Sensor and hand-collected datasets, on the other hand, are currently primarily available through a bewildering variety of formats and databases with no standardized metadata, quality control, or data organization. Our work to bring these sensor datasets into a common format with oceanographic cruise observations was a significant endeavor that responded to a real need and will improve our ability to map and understand the coastal ocean. Our priority for this synthesis was to bring nearshore and coastal datasets together, but we have included data extending well beyond the continental shelf to allow investigators some ability to examine biogeochemistry from offshore to nearshore environments even if this synthesis is not fully compatible with existing offshore gridded products.

Given upwelling- and respiration-driven low pH and low oxygen conditions manifest first in bottom waters, the way these conditions are explained in section 3.5 as within 50 m of the surface is confusing. It would be more intuitive to assess these conditions in the entire nearshore water column based on a bathymetric definition of nearshore, rather than defining nearshore as 50 km from the coast.

- Thank you, we really appreciate the improved data visualization suggestion. We have remade Figure 5 with data from within the 100m depth contour, which strikes a good balance of data availability, distance from shore (99% of the data is within 30 km of the mainland, with all "outlying" data associated with the Channel Islands in California), and ecological considerations of "nearshore" environments (e.g., environments where appreciable light still reaches the

benthos). Interestingly, the conclusions from this figure were very similar to those from the original figure since most data is coming from shallow, coastal moorings, but we agree that the bathymetric cutoff provides more intuitive support for these conclusions. Using a bathymetric cutoff for this figure also serves to show an additional way of interacting with the MOCHA dataset.

"Surface" and "near-surface" are used interchangeably, both defined in different parts of the manuscript as < 25 m. "Nearshore", "surface", and "near-surface" should all be defined early on in the results and used consistently throughout.

- We have replaced all "surface", "nearshore", and "near-surface" shorthand with explicit descriptions of the depth and spatial range.

The description (and potentially the application) of the secondary data quality control is inadequate. First, the original non-QC'd OOI data sets (section 2.3) need to be QC'd using recommended best practices specifically developed for OOI biogeochemical data sets (doi.org/10.25607/OBP-1865).

- Thank you, OOI published these recommended QA/QC practices just after our group worked with the data, so we were unaware of this publication. The OOI data was handled in close collaboration with OOI staff using code, best practices, and data cleaning techniques provided directly by them and now published in Palevsky et al., 2022. We have added references to these published OOI protocols to section 2.3 to clarify our actions around this data.

 Second, the description of the QC for the remaining data sets (section 2.4) sounds qualitative as written, as if the QC'er simply looked at property-property plots and flagged data points that looked bad. What the authors consider an "outlier" needs to be defined. Were outliers identified as a certain number of standard deviations of the linear (or some non-linear) relationship between parameters? What were the criteria for identifying "suspicious observations" (line 214)? Data QC routines need to be well documented and applied consistently throughout the data product using statistical analyses and thresholds to characterize quality.

- Thank you for this comment. In response to the first point, that QC practices should be objective and statistical, we generally agree but add some additional context. Most data pulled into this synthesis had been published and at least subject to automated QC processing. For these published datasets, our further

quality control role was akin to both the "human in the loop" and "comparisons among co-located data" steps of the OOI data's recommended best practices (Palevsky et al., 2022). Our secondary QC relied more on human judgment since automated QC practices are liable to miss clear instances of biofouling or significant sensor drift in automated sensors, as well as data that is unreasonable for the location, time, and depth while remaining within "normal" limits for the whole dataset. This is, by its nature, somewhat qualitative, but that is a response to the diversity of sampling schemes, observation frequencies, and habitats we were sourcing data from. The reliability of data associated with a tight time series dataset with samples every 20 minutes for 5 years is very different from a sporadic time series that includes data from three different seasons spaced across five different years, though both datasets can be effectively interpreted and quality-checked by a team with oceanographic expertise. In all cases, we opted towards data inclusion - flagging only data that was "unreasonable" or "unreliable" by the standards of that data set rather than a more aggressive stance. To clarify our QA/QC practices, we have added a flagging example to the Supplementary Information. This flagging example shows the raw data and previously published quality flags, the standard property-property and time series plots we used to double-check published data, and our changes and additions to the quality flags. All code and data associated with this example is fully available on our project Github repository (github.com/egkennedy/DSP_public_code).

It is also a best practice to state the constants used in carbonate chemistry calculations. In addition to Dickson et al. 2007, the authors should refer to more recent best practices for the use of constants in a broader range of temperature and salinity: doi.org/10.1016/j.marchem.2018.10.006; doi.org/10.1016/j.gca.2021.02.008; doi.org/10.1016/j.marchem.2014.07.004.

- Thank you for these reference suggestions. We realized that including calculated pH data in our manuscript provided little additional information and unnecessarily expanded our methods section, so we have removed all calculated pH data from our figures and discussion.

Lastly, the data products I am most familiar with all have a substantial acknowledgements section including funders of the observations, a long list of citations, and many coauthors because they include the major data providers in the

data product development. At minimum, Kennedy et al. should include all the data citations in the list of references. That requires referring to the metadata for each of the original data sets and including a data citation in the references if the data provider requests one be cited. I see citations provided in a table within NCEI Accession 0277984, but that is not trackable by the data providers. Those data citations are critical metrics that funders use to make decisions about what observational programs to support.

- We completely agree, and apologize that our misunderstanding of what could be included in the References section meant that we did not include dataset citations or DOIs in Table 1. We have since added those following an excellent example from Sutton et al., 2019 (https://doi.org/10.5194/essd-11-421-2019). Table 1 and the References section now both include full citations for each dataset. Now that the datasets can be identified by their citations, we have also shortened their titles and improved the overall readability of Table 1. The MOCHA dataset metadata table currently available on NCEI is also now included as a supplement.

**Minor comments:**

Line 44: Given this data product excludes seawater pH values derived using glass electrodes (for good reasons) they should consider referencing here the many other papers discussing coastal biogeochemical variability and change and not papers that utilize glass electrode data for estuarine and coastal pH monitoring. Many of the providers of the original data sets have published papers on this topic that could be cited instead.

- Thank you for pointing out this inconsistency. We have updated our references here to point towards higher quality pH monitoring efforts (lines 47-48).

Line 133: From my review of the original data sets, the product likely includes sensors using a membrane-based spectrophotometric method (the SAMI-CO2 as cited) and an equilibration-based method paired with an infrared gas analyzer (the MAPCO2). The phrase "autonomous equilibrium-based spectrophotometric pCO2 sensors" is a mix of the two.

- This has been corrected (lines 170-171).

Line 138: What is meant by "devices"? Are these sensors integrated into a CTD-rosette equipment package?

- Here, meant to reference all the sensors attached to a CTD-rosette, which might also include dissolved oxygen, pH, and chlorophyll sensors. This has been updated to "'CTD for observations from ship-side profiles with autonomous sensor arrays," (lines 178-179).

Table 1: Entry 52 title references the Cha Ba buoy, but the product only includes the cruise data for validating the buoy, but not the buoy data?

- That is correct. The Cha Ba buoy data will be included in future updates of this synthesis.

Lines 198-200: If the data product is going to propose and use a new set of flags, this section should explain why the authors chose to deviate from community-developed and widely-used standardized flagging schemes.

- We appreciate the suggestion to use a widely-known flagging scheme like QARTOD rather than developing a new scheme. Unfortunately, while some datasets we incorporated into this synthesis came with QARTOD flags, many others came with different quality information that was not easily mapped to the QARTOD scale. Additionally, what separates a QARTOD flag of 3 ("questionable/suspect") from a flag of 4 ("bad") depends on the project and investigator. Given the diversity of initial quality information available to us, we chose to use a simpler scheme of 1 = "plausible and reliable", 2 = "unevaluated", and 3 = "unreliable" to have an easily interpretable scale that we could map all datasets to. This reasoning has been added to section 2.4 (lines 257-259).

Section 2.5: Since nearshore biogeochemistry is heavily influenced by sub-daily processes, how do the authors account for potential bias in daily means when data are missing or flagged bad for a portion of the day?

- This was not a significant concern for this dataset for two reasons. For a given high resolution sensor, "unreliable" flags were applied either to individual outlying points or to multiday sections of data that showed extreme sensor drift and evidence of biofouling. In the former case, the removed data represents at most 1/24 of the day's information, making the loss of one measurement

minimally impactful. In the latter case, the period of sensor drift was identified by a date range within which all data was flagged, but there should be no days impacted significantly by flags applied only to a substantial portion of the day. There was also no evidence in our datasets for flags from the original data providers producing noticeable bias in this way. For close examinations of a given time series, we highly recommend data providers screen more closely for daily bias if necessary for their projects. In the context of data that is being compared to lower resolution sensors and daily, weekly, or seasonal discrete observations, the potential for bias in daily averages of a high resolution dataset is no more concerning than the practice of presenting discrete data observations as "the" oceanic conditions.

Top of page 21: First continued table entry looks incomplete.

- This table entry has been removed entirely since we no longer include calculated pH data.

Line 329: Figure 5 illustrates the ability of the data product to capture a monthly climatology. Seasonal variability could be interpreted as capturing all seasons over the entire time range of the data sets.

- Thank you for the clearer terminology. We have revised our discussion of Figure 5 to reference "monthly climatology" instead (lines 425-427).

Lines 335-337: Could differences in data density between those time periods be impacting this result?

- The data density is very comparable between the April to June and July to September periods in all regions, so the differences in variability and mean conditions between these two time periods are likely real.

Table 3 and associated discussion: I was surprised to not see a comparison to, or at minimum a mention of, previously-published TA-S relationships for these regions.

- We have substantially revised this section following the discovery of a years-old quality-control issue with some autotitrator data supplied by the author team (new text in lines 485-516). The data impacted have been removed from the MOCHA compilation and this manuscript. As such, we have replaced the old Table 3 and

Figure 7 with an updated Figure 7 that shows the regional and nearshore (< 2 km distance) and offshore TA-S relationships. We now discuss these results in context with Fassbender et al. (2017), Cullison Gray et al., (2011) and others.

Line 419: "Saildrone" is a company. These types of oceanographic platforms are commonly called Uncrewed Surface Vehicles (USVs).

- This has been fixed.

Line 418: This product should be named and cited.

- We now explicitly name and cite SOCAT here (lines 565-566).

Line 423: The mention of considering deeper water here is confusing because there is no mention of a desire to assess bottom water earlier in the manuscript. The analysis is focused on varying definitions of surface water.

- We have made it clearer throughout the manuscript that our case examples represent just a few of the potential uses we envision for the MOCHA synthesis. While we focus primarily on surface water for our case examples in order to highlight the coastal autonomous sensors and moorings this compilation includes, we have taken care to ensure that the MOCHA synthesis is also supportive of investigations focused on deeper waters.

---

## Author Response (AR2)

We greatly appreciate the time that Reviewers 1 and 2 took to read and comment on our revised manuscript. As Reviewer 2 had no further comments, we only include Reviewer 1's comments below. Our point by point responses are bulleted just below the relevant comment. Except as noted, we have made all requested revisions to the manuscript as well as checked it for clarity and typos.

**Reviewer 1 Comments on the Revised Manuscript**
I appreciate the authors' thoughtful and thorough responses to all reviewers' comments. I am satisfied with how they addressed my substantive comments and am convinced by their replies where they did not agree with me. I feel that the result is an article that much more clearly justifies and differentiates itself relative to other major data compilation efforts (e.g., SOCAT, GLODAP). This will be a valuable information product for many end users.

- Thank you for the kind words. Our manuscript benefited significantly from all three reviewers' feedback and we are pleased to hear that the revised manuscript is much stronger. We appreciate your time and willingness to do a close read of the revised manuscript.

However, I have a few remaining concerns, most of them quite basic:

1) Lines 209-213—It would be helpful to spell out which constants they used with seacarb for readers' understanding. If this is elsewhere, no need to repeat, but in the two places I noticed this information, the various constants were not identified, as is standard practice for CO2 system calculations.

- Thank you for the reminder. We have added the constants we used to our methods section discussion of converting pH measurements to *in-situ* conditions for discrete samples in Section 2.2. As there are no longer any calculated carbonate system parameters in this dataset or manuscript (in contrast to Version 1), this is the only step in which they are used.

2) In at least two places, they said "principle carbonate system parameters"—I think they mean "principal," although I'm more used to thinking of it as two of the "measurable" CO2 system parameters.

- You are correct; we did mean "principal carbonate parameters" rather than "principle carbonate parameters." We have corrected this in the revised manuscript. We prefer "principal" over "measureable" in this case because, while

measuring any two of the parameters pH, total alkalinity, dissolved inorganic carbon, and $pCO_2$ will allow you to calculate the full carbonate system, there are other additional components of the carbonate system that can be measured that do not provide this benefit (e.g., the concentration of $CO_2^{2-}$ ions).

3) They added stars to Figure 3 to illustrate where the locations referred to in the caption are, which is an improvement. It would be great to add them to Figure 1 too, for people like me who like to see things on an actual map.

- We appreciate this suggestion to improve map readability, but ultimately found that it made the figure more cluttered and hard to read, especially given that the state borders are already part of the figure. We have added a light gray background to the land and darkened the state lines to heighten the contrast in the figure and make it easier for readers to orient themselves, however.

4) Table 1—I really appreciate the authors adding citations (and the fair data use statement at the end) to the table. However, I think it's unfortunate to have removed the parameters included in each data set in the table. Could that information be restored? I also think some indication of the intrinsic uncertainty of the various input data sets would be very helpful for end users to have. Would it be possible to go through and flag the data sets in the table with respect to overall uncertainty? I realize this may differ by parameter and is not a small amount of work if they don't already have it somewhere (though I would think they do). If this information is elsewhere (e.g. in the detailed metadata), a simple statement clearly indicating where this info is should suffice for pointing readers to it. I am even thinking a binary indicator added to the table may be adequate—such as a superscript 1 indicates climate-quality vs superscript 2 indicates weather-quality data (where the superscript could either go on the ID column or by parameter if those can be added back in). Information on what I mean by climate vs weather can be found in the GOA-ON principles (Newton et al. 2015), at least for the OA(H?) parameters, if not also for T and S.

- Thank you for these suggestions. We have added back in the "parameters" column as suggested. We appreciate the suggestions of how to add an indication of data quality to the datasets table or supplemental table. However, this is challenging to determine for most datasets in our synthesis, as many datasets are a mix of both data types. As you note, the data quality varies by parameter. It also frequently varies through time within the same dataset. As such, we felt that this

determination was beyond the scope of the manuscript. To alert readers and users to this, we have added a caution about data quality to two places within the paper. In the caption of Table 1, we added "Users need to be mindful of the difference between climate-quality and weather-quality datasets and assess the suitability of these datasets for their needs (Newton et al., 2015)." Secondly, we added the following sentences to the beginning of our "Quality Control" section in methods (Sect. 2.4): "This quality standardization did not extend to raising all datasets to a "climate-quality" standard (Newton et al., 2015). Users of these data should be aware of the difference between climate-quality versus weather-quality data, as both types of data are included in this synthesis and often coexist within the same datasets."

5) I like the rewritten TA-S section and the new figure. Much stronger.

- This comment is much appreciated. We are grateful for this and another reviewer's comments that pushed us to substantially rework this section.

6) There are at least a few places where the word "data," which is always plural, has a singular verb conjugation.

- Thank you for this reminder. We have found and fixed these instances.

7) I've selected data quality "good" above because I am not sure ALL the data included are "climate-quality" (which I'd equate to "excellent", vs. "good" for weather-quality data).

- This is correct and feels like a fair assessment; we do have some weather-quality datasets in our synthesis. For some applications, this will likely increase the utility of the synthesis product since it significantly increases the data available. However, including these data sets comes at the cost of requiring investigators to filter the synthesis product and carefully screen data sources if their work requires only climate-quality data.

8) Finally, I couldn't get all the way through the supplement because some of the figures were bogging my system down (despite having had larger graphics files open earlier today in Acrobat with no problem).

- We have fixed the issue with the way our figures were embedded in the supplement so it should now open without difficulty. We have also taken this

opportunity to add descriptive figure captions and make some minor formatting adjustments to the supplement.

Overall, well-done synthesis and well-written paper. I'll be excited to see this published.